# Prevalence of *Salmonella* spp. and *Escherichia coli* in the feces of free-roaming wildlife throughout South Korea

**Rahman M. Mafizur**[1,2], **Lim Sangjin**[1,3], **Park Y. Chul**[1]*

**1** Division of Forest Science, Kangwon National University, Chuncheon, Republic of Korea, **2** Department of Biotechnology and Genetic Engineering, Islamic University, Kushtia, Bangladesh, **3** Institute of Forest Science, Kangwon National University, Chuncheon, Republic of Korea

* parky@kangwon.ac.kr

**Data Availability Statement:** All data are available in the paper. The accession numbers for NCBI GenBank are ON205852- ON205944.

## Abstract

Wildlife can carry pathogenic organisms, including viruses, bacteria, parasites, and fungi, which can spread to humans and cause mild to serious illnesses and even death. Spreading through animal feces, these pathogens significantly contributes to the global burden of human diseases. Therefore, the present study investigated the prevalence of zoonotic bacterial pathogens, such as *Salmonella* spp., *Escherichia coli*, and Shiga toxin-producing *E. coli* (STEC), in animal feces. Between September 2015 and August 2017, 699 wildlife fecal samples were collected from various agricultural production regions and mountainous areas in South Korea. Fecal samples were collected from wild mammals (85.26%, 596/699) and birds (14.73%, 103/699). *Salmonella* spp. and *E. coli* were present in 3% (21/699) and 45.63% (319/699) of the samples, respectively. Moreover, virulence genes *stx1* and both *stx1* and *stx2* were detected in 13.30% (93/699) and 0.72% (5/699) of the samples, respectively. The 21 *Salmonella* spp. were detected in badgers (*n* = 5), leopard cats (*n* = 7), wild boars (*n* = 2), and magpies (*n* = 7); STEC was detected in roe deer, water deer, mice, and wild boars. Through phylogenetic and gene-network analyses, the *Salmonella* spp. isolates (*n* = 21 laboratory isolates, at least one isolate from each *Salmonella*-positive animal fecal sample, and *n* = 6 widely prevalent reference *Salmonella* serovars) were grouped into two major lineages: *S. enterica* subsp. *enterica* and *S. enterica* subsp. diarizonae. Similarly, 93 *E. coli* isolates belonged to *stx1*, including three major lineages (groups 1–3), and *stx1* and *stx2* detected groups. To the best of our knowledge, this is the first report of a wild leopard cat serving as a reservoir for *Salmonella* spp. in South Korea. The research findings can help manage the potential risk of wildlife contamination and improve precautionary measures to protect public health.

## Introduction

Wild animals are competent asymptomatic reservoirs of *Salmonella* spp. and Shiga toxin-producing *Escherichia coli* (STEC), which are the most commonly reported causes of bacterial

**Funding:** This work was supported by Cooperative Research Program for Agricultural Science & Technology Development (Project No. PJ0108592016), Rural Administration, Republic of Korea and Korea Institute of Planning and Evaluation for Technology in Food, Agriculture and Forestry (Project No. 122013-2). The funders had no role in study design, data collection and analysis, decision to publish, or preparation of the manuscript.

**Competing interests:** The authors have declared that no competing interests exist.

food-borne zoonotic diseases in South Korea [1–5]. *Salmonella* spp. are found in the gastrointestinal tracts of both domestic [6] and wild animals [7], as well as in humans [8]. They are transmitted through feces into agricultural produce [9], particularly leafy vegetables, during irrigation, cultivation, and processing [10], and they can spread from one animal to another (domestic to wild and vice versa) and from one person to another. Moreover, they are a common food-borne pathogen, accounting for 6.3% of *E. coli* and 3.5% of *Salmonella* infections in Korea [11]. *Salmonella* can survive outside the body for years in a variety of environments, including soil and sludge, where temperature and nutrient availability vary [12–15]. *E. coli* O157:H7 is the predominant STEC serotype that causes life-threatening hemorrhagic colitis and hemolytic uremic syndrome [16], whereas *E. coli* O157 infections in humans result in clinical syndromes ranging from asymptomatic to severe [17].

Zoonotic pathogens such as *Salmonella* spp. and STEC have received special attention owing to their high prevalence and host ubiquity [18]. Agricultural produce areas can be contaminated by pathogens from raw and poorly composted manure, contaminated irrigation water, and wildlife fecal deposition [18–20]. Pathogens are found in the gut and feces of warm-blooded animals, such as livestock, wildlife, poultry, and birds [21–24]. Irrigation water contaminated with feces is often responsible for the pathogen contamination of leafy greens [25, 26]. *Salmonella* was detected in the feces of various animals, including wild boar [27, 28], wild foxes, ducks [29], stray dogs, coyotes [30], hedgehogs [31], tigers [32], cattle [33], magpies [31], and waterfowls [34]. Owing to their omnivorous feeding habits, wild animals are major reservoirs of *Salmonella* and can be exposed to *Salmonella* colonization through fecal shedding [31]. Consequently, they can act as spreaders, contaminate agricultural products and animals, and cause human infections. Owing to their increasing interactions, humans, domestic animals, and wildlife can all be carriers of zoonotic pathogens [31, 34].

Several salmonellosis outbreaks caused by *Salmonella* have been reported, and the number of cases has increased [11] over the last five years (2015–2019) in South Korea [35–39]. *Salmonella* is responsible for 7.7% of food-borne illnesses in Korea, according to Gwack et al. (2010). Between 1998 and 2007, Kim (2010) identified *Salmonella* in nearly 10,000 patients suffering from food- and water-borne illnesses [40]. Furthermore, the majority food-borne illnesses (approximately 340,000 cases per year) occurred between 2008 and 2012 [41]. According to Korea Centers for Disease Control and Prevention (KCDC), an increasing trend in food-borne and water-borne diseases was observed in South Korea from 2015 to 2019, with the causative pathogenic agents, being *E. coli* for 6.3% of the cases, and *Salmonella* spp. for 3.5%.

Animals such as cattle and pigs must be treated with precaution because they are considered a most frequent sources of salmonellosis outbreaks in humans in Korea [27, 36, 42]. Ultimately, the cross-contamination of fresh fruits and vegetables is caused by different pathogenic agents, including *Salmonella* and *E. coli* in Korea [43].

Pathogenic *E. coli* strains have been detected in the feces of multiple wildlife species, including deer [44], feral swine [45], wild sheep [46], wild boar [47, 48], wild goats [49], waterfowl [34], and ungulates [50, 51]. Several recent outbreaks have suggested that wild animals are *E. coli* O157 reservoirs in agricultural production areas and that they contaminate fresh produce in the field either directly or through cross-contaminated agricultural water [26, 52–54].

This study aimed to isolate and identify pathogenic *E. coli* and *Salmonella* spp. from wildlife and bird feces using a common primary and secondary enrichment step, followed by selective cultural methods, biochemical and serological latex agglutination tests, molecular markers for polymerase chain reaction (PCR) amplification, and phylogenetic and gene network analysis.

## Materials and methods

### Sample collection and processing

Wild animal feces were collected from multiple agricultural regions near the mountain forests of 17 counties in Gangwon Province (S1 Fig and Table 1) and from 10 counties in other provincial regions within each of the five geographical regions (northeastern, northwestern, middle-eastern, central, and southeastern) across South Korea (Fig 1 and S1 Table). We collected samples at least three times a month from March to November, spanning three seasons (spring, summer, and autumn) each year from 2015 to 2017. No fecal samples were collected during winter (December to February) in any year (S2 Table). Based on sample size calculations (Glenn 2002; http://www.winepi.net/uk/index.htm), the considered confidence level was 99%, population size unknown, expected standard deviation 0.5, accepted absolute error d = 0.005, and final sample size was 664. We then aimed to collect approximately 700 samples within three seasons: spring (March–May), summer (June–August), and autumn (September–November). More than double samples were collected in the summer season compared to the other two seasons because of the possibility of pathogen prevalence of *Salmonella* and STEC, as previously described [55]. Fresh fecal samples were collected from leafy green vegetable fields or nearby regions, where fresh fecal pools of wild animals were found, and our laboratory routinely surveyed the representative survey area for sample collection (Seoraksan National park, Sokcho, Gangwon-do), by a combination of tracking feces and camera traps [56]. Fresh surface-deposited feces from each individual were stored in a sterile 50 ml Falcon tube (Fisher Scientific Supply Co., USA) or a clean vinyl zipper bag with the help of a disposable wooden stick. To ensure fresh fecal material, the sites were surveyed in the afternoon of the day prior to the sampling day and then revisited the following morning to collect fresh feces. Rodent fecal samples were collected from live rodents captured using Sherman traps [57]. The GPS coordinates of each fecal sample collection point were recorded using a 60CSx (Garmin Inc., USA). The fecal samples were transported to the laboratory in ice boxes on the day of collection and cultured in the laboratory on the day of arrival within approximately 24 h. To identify the collected wild animal feces, we carefully determined the size (measuring the size of the collected feces with a ruler on the sampling date), shape, and color of the feces (S2 Fig) and ruptured the fecal material to check the composition (https://www.discoverwildlife. com/how-to/identify-wildlife/how-to-identify-animal-droppings/, accessed on May 26, 2022). We routinely surveyed the fields for animal intrusions [58–60] and were trained to identify wild animals, including wildlife common in agricultural areas (S3 Fig). Moreover, we confirmed the presence of wild animals in the study area using trail cameras (Browning, BTC-5PXD, USA), see the re-presentative sample collection and trial camera set-up survey area of Odaesan National Park (Pyeongchang, Gangwon-do) (S4 Fig).

### Culture and detection of *E. coli*

*E. coli*, STEC, and other bacterial species have been detected in fecal samples using a modified culture method [15, 61, 62]. First, fecal samples were cultured in a final volume of 10 mL (diluted 1:10) with non-selective buffered peptone water (BPW) overnight at 37˚C. Then, 10μL of enrichment broth was streaked with a loop onto *E. coli*-selective eosin methylene blue (EMB) agar medium and STEC-selective cefixime tellurite sorbitol MacConkey (CT-SMAC) agar medium (Oxoid, UK) and incubated at 37˚C for 24–48 h. The plates were examined for colony forming units (CFU) and sub-cultivated on an EMB to collect pure colonies. Colonies of *E. coli* and STEC were presumably confirmed based on the basis of colony morphology [63, 64].

**Table 1. Prevalence and distribution pattern of *Salmonella* and *Escherichia coli* in different counties across South Korea.**

| Collection locality | Collection locality ID | No. of fecal samples | Percentage (No. of *Salmonella*-positive sample) | Percentage (No. of *E. coli*) | STEC[g] | |
|---|---|---|---|---|---|---|
| | | | | | *stx1*PCR[f] | *stx1+stx2*[e] PCR |
| Gangneung, Gangwon-do | GRG | 6 | 0 (0) | 66.67 (4) | 0 (0) | 0 (0) |
| Goseong, Gangwon-do | GSG | 23 | 0 (0) | 52.17 (12) | 43.48 (10) | 0 (0) |
| Samcheok, Gangwon-do | SCG | 55 | 5.45 (3) | 25.45 (14) | 16.36 (9) | 0 (0) |
| Sokcho, Gangwon-do | SKG | 13 | 0 (0) | 46.15 (6) | 38.46 (5) | 0 (0) |
| Yanggu, Gangwon-do | YGG | 16 | 0 (0) | 56.25 (9) | 0 (0) | 0 (0) |
| Yangyang, Gangwon-do | YYG | 17 | 0 (0) | 64.71 (11) | 35.29 (6) | 0 (0) |
| Yeongwol, Gangwon-do | YWG | 74 | 0 (0) | 21.62 (16) | 9.72 (7) | 1.39 (1) |
| Wonju, Gangwon-do | WJG | 5 | 0 (0) | 60.00 (3) | 20.00 (1) | 20.00 (1) |
| Inje, Gangwon-do | IJG | 13 | 0 (0) | 38.46 (5) | 7.69 (1) | 0 (0) |
| Jeongseon,Gangwon-do | JSG | 4 | 0 (0) | 75 (3) | 0 (0) | 0 (0) |
| Cheolwon, Gangwon-do | CWG | 42 | 0 (0) | 30.95 (13) | 4.76 (2) | 0 (0) |
| Chuncheon, Gangwon-do | CCG | 90 | 4.44 (4) | 41.11 (37) | 23.33 (21) | 1.11 (1) |
| Taebaek, Gangwon-do | TBG | 44 | 2.27 (1) | 47.73 (21) | 18.18 (8) | 0 (0) |
| Pyeongchang, Gangwon-do | PCG | 47 | 0 (0) | 31.91 (15) | 4.26 (2) | 0 (0) |
| Hongcheon, Gangwon-do | HNG | 18 | 0 (0) | 33.33 (6) | 5.56 (1) | 0 (0) |
| Hwacheon, Gangwon-do | HCG | 44 | 0 (0) | 38.36 (16) | 2.27 (1) | 0 (0) |
| Hoengseong, Gangwon-do | HSG | 5 | 0 (0) | 0 (0) | 0 (0) | 0 (0) |
| Seoul city | HKS | 31 | 6.45 (2) | 54.84 (17) | 6.45 (2) | 0 (0) |
| Geochang, Gyeongsangnam-do | GNG | 23 | 0 (0) | 73.91 (17) | 21.74 (5) | 8.70 (2) |
| Busan city | BUS | 39 | 25.64 (10) | 94.87 (37) | 2.56 (1) | 0 (0) |
| Gure, Gyeongsangnam-do | GYG | 8 | 0 (0) | 75 (6) | 25.00 (2) | 0 (0) |
| Yeongju, Gyeongsangbuk-do | YOG | 30 | 0 (0) | 43.33 (13) | 11.54 (3) | 0 (0) |
| Yeongcheon, Gyeongsangbuk-do | YNG | 13 | 7.69 (1) | 61.54 (8) | 7.69 (1) | 0 (0) |
| Bonghwa, Gyong-buk-do | BHG | 26 | 0 (0) | 63.33 (19) | 13.33 (4) | 0 (0) |
| Cheongyanggun, Chungchang nam-do | CHN | 6 | 0 (0) | 83.33 (5) | 0 (0) | 0 (0) |
| Jinan, Jeollabuk-do | JLD | 7 | 0 (0) | 85.71 (6) | 14.29 (1) | 0 (0) |
| | **Total** | **699** | **3.00 (21)** | **45.64 (319)** | **13.30 (93)** | **0.72 (5)** |

[e] "stx1+ stx2 PCR" = *Escherichia coli* was detected by both primers of the Shiga toxin gene; PCR = Polymerase Chain Reaction

[f] "stx1PCR" = *Escherichia coli* was detected by one Shiga toxin gene

[g] 'STEC' = Shiga toxin-producing *E. coli*

## Culture and detection of *Salmonella*

*Salmonella* was detected by incubating the samples in a non-selective primary enrichment (BPW) medium at 37˚C for 18–24 h. Subsequently, 1mL of the enrichment broth culture was added to 9 mL of selective Muller-Kauffmann tetrathionate enrichment broth (Difco, Becton Dickinson, USA) and incubated for 24 h, and 100 μL of primary enrichment broth culture was incubated in 10 mL of Rappaport-Vassiliadis (RV) enrichment broth (Oxoid, UK) for 24 h. After enrichment, 10 μL of each sample was streaked on *Salmonella*-selective *Salmonella*-Shigella (SS), xylose lactose tergitol™ 4 (XLT4), and Hektoen enteric (HE) agar media (Difco, Becton Dickinson, USA) and incubated at 37˚C for 24–48 h. The plates were then examined for *Salmonella* spp. Colony forming units use colony morphology and biochemical test (USFDA, 1995). Suspected *Salmonella* isolates were re-examined using the same media, and the bacterial

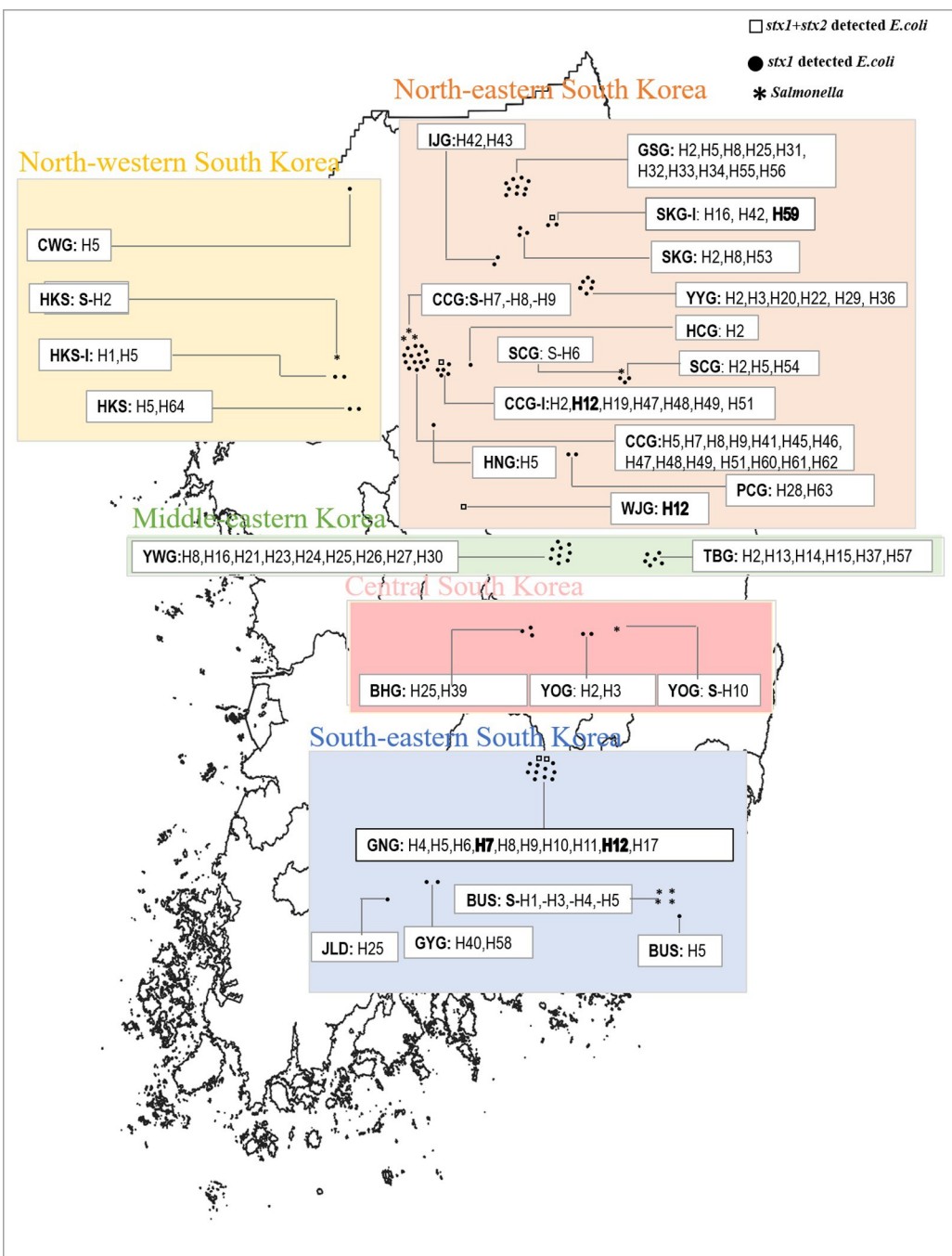

**Fig 1. Distribution map of bacterial-borne zoonotic pathogen Shiga toxin genes (*stx1* and *stx1+stx2*)-detected in *E. coli*, and *Salmonella* from the feces of wild mammals and birds in South Korea (2015–2017).** The locality abbreviations with haplotypes are presented in parenthesis. The locality abbreviations are as follows: Goseong, Gangwon-do (GSG); Samcheok, Gangwon-do (SCG); Sokcho, Gangwon-do (SKG); Yangyang, Gangwon-do (YYG); Yeongwol, Gangwon-do (YWG); Wonju, Gangwon-do (WJG); Inje, Gangwon-do (IJG); Cheolwon, Gangwon-do (CWG); Chuncheon, Gangwon-do (CCW); Daeryongsan, Chuncheon, Gangwon-do (CCW-I); Taebaek, Gangwon-do (TBG); Pyeongchang, Gangwon-do (PCG); Hongcheon, Gangwon-do (HNG); Hwacheon, Gangwon-do (HCG); Seoul city (HKS); Geochang, Gyeongsangnam-do (GNG); Busan city (BUS); Gure, Gyeongsangnam-do (GYG); Yeongju, Gyeongsangbuk-do (YOG); Yeongcheon, Gyeongsangbuk-do (YNG); Bonghwa, Gyong-buk-do (BHG); Jinan, Jeollabuk-do (JLD). 'Location codes: S-H (no. of haplotypes) means *Salmonella* haplotypes and '*' indicates *Salmonella* presence, and '°' indicates the *stx-1*-positive *E. coli* presence in the respective locality/region. For the illustrative purposes, we used a frame of map of the South Korea from the Korea National Spatial Data Infrastructure Portal (http://www.nsdi.go.kr) that was created with Quantum Geographic Information System v3.2 (QGIS v3.2) software. Geographical directions were shaded with a specific color for the north-western (cream), north-eastern (peach), middle-eastern (light-green), central (salmon), and south-eastern (lavender) directions.

isolates produced black-centered colonies. The triple sugar iron procedure (TSI: three sugars, lactose, sucrose, and glucose) was used to further test *Salmonella* colonies. A well-isolated presumptive colony was stabbed through the center and streaked onto the surface of the agar slant using a sterilized straight inoculation needle. *Salmonella*-positive colonies produced black slant (upper part of the tube) and red butt (lower part of the tube) containing $H_2S$ gas. Presumptive *Salmonella* colonies were tested using polyvalent antibodies. A latex agglutination polyvalent test kit was used for presumptive identification of *Salmonella* spp (Oxoid, UK).

### Extraction of total genomic DNA and PCR amplification

The bacterial colonies identified by morphology on selective media were investigated by PCR using molecular markers. The colonies were streaked onto nutrient agar media and a single colony was collected using a sterilized toothpick. The colonies were incubated at 35˚C for 18 h in 5 mL of lactose broth solution. Genomic DNA was extracted from 1 mL of lactose broth culture fluid using a DNeasy Blood and Tissue Kit, according to the manufacturer's instructions (Qiagen, Valencia, CA, USA).

*E. coli* colonies on EMB agar were identified by PCR amplification of a hypervariable region (HVR) of the bacterial 16S rRNA gene, using the HVR primer set [65] PCR amplification was performed using a final 25 μL reaction volume containing 10 mM Tris–HCl (pH 8.4), 50 mM KCl, 4 mM $MgCl_2$, 200 mM of each dNTP, 50 pmol of each primer, 2 U ExTaq polymerase, and 1 μL of genomic DNA.

The STEC colonies on CT-SMAC agar were molecularly identified by PCR amplification of Shiga toxin genes (*stx1* or *stx2*) using the *stx1* or *stx2* primer set [66]. *E. coli* O157:H7 was identified using a single nucleotide polymorphism (SNP)-based marker [67] The STEC-positive colonies were amplified and sequenced with the developed SNP-encompassing-based forward primer (TGACTCGCCGTATGTGCC) and reverse primer (TGCCCGGCAGATAAGTGC) according to our previous study [68]. The following sequence accession numbers were submitted to GenBank (ON205852–ON205944), and detailed sample information is provided in S3 Table.

*Salmonella* colonies on the SS, HE, and XLT4 plates were identified by amplifying the *invA* or *iroB* genes, which are shared by all *Salmonella* species, using PCR with the invA or iroB primer sets [69, 70]. The positive control strain of *S. enterica* subsp. *enterica* serovar Enteritidis (NCCP 14545) and a negative control of *E. coli* (NCCP14034) were used to confirm *Salmonella* amplification. PCR was conducted using the following reaction conditions: an initial denaturation for 5 min at 94˚C, followed by 35 cycles of denaturation for 1 min at 94˚C, annealing at 55˚C (HVR, *stx1*, and iroB primer), 52˚C (*stx2*), and 62˚C (*invA*) for 60 s, extension for 1 min at 72˚C, and a final extension of 10 min at 72˚C. The purified PCR products were subjected to electrophoresis on a 1.0% agarose gel and purified using a DNA gel extraction kit (Qiagen, Valencia, USA). CA, USA) see **S5A and S5B Fig**.

*Salmonella* serovars were identified and matched using our SNP-based marker and gene sequencing. The 21 *Salmonella*-positive samples were amplified and sequenced using the developed SNP-based forward primer (GGCGTTGAAGAAGCAGCG) and reverse primer (ACGGCCTACCCAGGTGAT), according to our previous study [67]. The following sequence accession numbers were submitted to GenBank (*Salmonella* accession number: OM793284–OM793304). SNP-nucleotide-targeted primers (SNP-encompassing primer S6 **Fig**) based on the partial sequences of the penicillin-binding protein gene (*mrcB*), widely known for antibiotic resistance (β-lactam antibiotic resistance due to point mutation of the *mrcB* gene [71]. We used this primer to sequence *Salmonella* isolates from the feces of wild animals. A schematic diagram of SNP-encompassing primers (mrcB-9-Sbon-F/R) is presented in S6 **Fig**. In addition, PCR amplification was performed using 25 μL reaction mixture containing 10 mM Tris-

HCl (pH 8.4), 50 mM KCl, 4 mM MgCl2, 200 mM of each dNTP, 50 pmol of each primer, 2 U ExTaq polymerase, and 1 μL (5 ng/μL) of genomic DNA, and the final volume was adjusted with distilled water. The PCR reaction was conducted at 95˚C for 5 min, followed by 35 cycles of denaturation for 35 s at 95˚C, annealing at 55˚C for 30 s, extension at 72˚C for 1 min 30 s, and a final elongation at 72˚C for 5 min. The PCR products were separated using electrophoresis with a 2% agarose gel and were purified using a Gel & PCR Purification Kit (Biomedic Co., Ltd., Seoul, South Korea) and sequenced using a BigDye Terminator v3.1 Cycle Sequencing Kit (Applied Biosystems, Foster City, CA, USA) and an ABI 3730 DNA Analyzer (Applied Biosystems, Foster City, CA, USA). All *Salmonella* sequences were compared with those in the NCBI BLAST database (maximum score, coverage, identity, and E-value; checked with the naked eye). The *Salmonella* serovar was considered to align the reference serovar sequences (six reference sequences) with the topmost matched *Salmonella* strains (highest similarity, coverage, and E-value) and SNP-based multiplex marker [67].

## Phylogenetic analysis of isolated *E. coli* and *Salmonella* strains

The obtained *Salmonella* spp. sequences were compared with homologous sequences deposited in GenBank using BLASTN2.2.31+ [72]. The genetic sequence data were deposited into the NCBI database with GenBank accession numbers OM793284 - OM793310.

We used 21 isolates from 21 *Salmonella*-positive animal fecal samples for phylogenetic and gene network analyses (median joining network, [MJN]). The six widely prevalent reference *Salmonella* serovars [*S.* Abony (BA1800061), *S.* Enteritidis (NCCP-14545), *S.* Agona (NCCP-12231), *S.* Typhimurium (NCCP-14760), *S.* Typhi (NCCP-14641), and *S. enterica* (NCCP-15756)] were used to build a phylogenetic relationship, which was inferred using neighbor-joining (NJ) analyses implemented in MEGA 10.0.14. The confidence of the branches in the maximum likelihood (ML) trees was assessed using bootstrapping searches with 1000 replicates. A NJ tree was inferred using Kimura's 2-parameter model, with bootstrapping investigations of 1000 replicates.

Similarly, 93 isolate sequences (few of the isolates that failed to recover fresh sequences) were picked from 93 *stx1* positive animal fecal samples for phylogenetic analysis, which were used for NJ and ML analyses, with bootstrapping investigations of 1000 replicates.

## Network analysis of isolated and *E. coli* and *Salmonella* strains

A widely used network approach for visualizing and understanding genetic relationships (for analyzing and visualizing with different color codes) between the *ileS* partial gene sequences of 93 *E. coli* isolates was constructed using the MJN approach with default parameters in the NETWORK software (version 4.6.1.2, Fluxus Technology Ltd., Suffolk, UK). *Stx1*-positive *E. coli* isolates were constructed using the MJN approach with default parameters in NETWORK software (version 4.6.1.2, Fluxus Technology Ltd., Suffolk, UK). The analysis was aimed at constructing haplotype networks of the partial *ileS* sequences [68], with color-coded haplotypes for animal hosts and geographical origins. Haplotype diversity and nucleotide diversity of *stx1* detected *E. coli* were calculated using DnaSP.5.10 [73].

The partial gene sequences of penicillin binding protein (*mrcB*) (amplicon length = 763 bp, refer to S6 Fig) of 21 isolated *Salmonella* strains and widely prevalent 6 reference serovar strains were sequenced in this study [(accession number of reference strains: Abony (OM793305), Typhimurim (OM793306), Enteritidis (OM793307), Agona (OM793308), enterica (OM793309), and Typhi (OM793310)]. The NCBI-acquired *Salmonella* serovar strains were used to perform an MJN analysis with default parameters in NETWORK software (version 4.6.1.2, Fluxus Technology Ltd, Suffolk, UK). The analysis was aimed at constructing haplotype

networks of partial *mrcB* sequences, with color-coded haplotypes of animal host (*Salmonella* is found in wild animal feces) and geographical origins (different places in South Korea). Haplotype and nucleotide diversities of *Salmonella* were determined using DnaSP.5.10 [73].

### Ethics information

The collection of samples was done with the permission of a recognized authority in South Korea. Specifically, rodent sample collection was conducted with permission from the local government and Korea National Park Research Institute (approval number: 778), according to the guidelines of the government and institute. In addition, wildlife fecal samples were collected from the ground near agricultural areas, and no animals were handled other than rodents. Live rodents were captured and released with a rodent trap to collect fecal samples. For this, we captured rodents in Sherman traps. We release the rodents in their natural habitat after collecting fecal generally within three hours (no slaughter).

### Statistical analysis

Regression analysis was used to determine whether a sample was positive or negative for *Salmonella* and *E. coli*. To determine the factors associated with the prevalence of the tested pathogenic agents (*Salmonella*, *E. coli*, *stx1*-detected *E. coli*, *stx1*+*stx2*-positive *E. coli*), we conducted regression analysis (a method to determine the reason-result relationship of independent variables with dependable variables) for each animal, season, and collection site. The binary/ dichotomous variable (yes/no) indicated whether any pathogen was detected in a fecal sample. The regression analysis outcome was coded on a continuous scale (0 or 1, where "1" represents the outcome of tested pathogens and "0" indicates absence in the outcome of tested pathogens). The tested pathogenic agents could be influenced by the seasons (summer, spring, and fall) and diverse categories of wild animals. Pathogens (each category of pathogens, such as *E. coli*, *stx1*-detected *E. coli*, *stx1*+*stx1*-detected *E. coli*, and *Salmonella*) were considered as dependent variables and season (autumn, summer, and spring) and animals (13 categories of wild animals) were considered as co-variable data.

To identify the factors associated with the prevalence of pathogenic agents at each sampling or collection site. The information was entered into Microsoft Excel 2013 spreadsheets and exported for analysis using the Statistical Package for the Social Sciences (SPSS 24). The chi-squared test was used to assess the validity of the model. Following model estimation, the odds ratio and 95% confidence interval (CI) were derived. A *P*-value ≤0.05 was used to detect significantly associated factors for pathogen presence in the logistic regression analysis. The percentages displayed in the results are the number of positive or negative samples divided by the total number of samples obtained from that subset of data.

## Results

### Sample collection

Between September 2015 and August 2017, 596 fecal samples from 12 mammalian species and 103 fecal samples from a single bird species (*Pica sericea*) were collected from multiple agricultural regions across South Korea (**Table 2**). The largest number of fecal samples was collected from water deer (302 samples), followed by magpies (103 samples) and leopard cats (98 samples), whereas fecal samples from the least weasel (three samples) and long-tailed goral (four samples) were rarely collected (Table 2).

**Table 2. Prevalence of bacteria-borne zoonotic pathogens *Salmonella*, *E. coli*, and Shiga toxin genes (*stx1* and *stx1+stx2*) in the feces of wild animal species.**

| Species (scientific name) | No. of fecal samples (n) | Percentage of pathogen-positive sample (No. pathogen-positive sample/No. of samples tested) | | | | | | |
| --- | --- | --- | --- | --- | --- | --- | --- |
| | | *Salmonella* | *E. coli*[C] | | | STEC | | |
| | | | EMB | TSI[e] | 16S PCR | CT-SMAC | *stx-1* PCR | *stx-1 + stx-2*[d] PCR |
| Water dear (*Hydropotes inermis*) | 302 | 0 (0/302) | 35.43 (107/302) | 35.43 (107/302) | 35.43 (107/302) | 8.94 (27/302) | 8.94 (27/302) | 0.33 (1/302) |
| Magpie (*Pica sericea*) | 103 | 6.8 (7/103) | 53.40 (55/103) | 53.40 (55/103) * | 53.40 (55/103) | 12.62 (13/103) | 12.62 (13/103) | 0 (0/103) |
| Leopard cat (*Prionailurus bengalensis*) | 98 | 7.14 (7/98) | 69.39 (68/98) | 69.39 (68/98) | 69.39 (68/98) | 20.41 (15/98) | 20.41 (15/98) | 0 (0/98) |
| Striped field mouse (*Apodemus agrarius*) | 55 | 0 (0/55) | 40 (22/55) | 40 (22/55) | 40 (22/55) | 21.81 (12/55) | 21.81 (12/55) | 3.63 (2/55) |
| Wild boar (*Sus scrofa*) | 40 | 5 (2/40) | 55 (22/40) | 55 (22/40) | 55 (22/40) | 7.5 (3/40) | 7.5 (3/40) | 2.5 (1/40) |
| Roe dear (*Capreolus capreolus*) | 24 | 0 (0/24) | 66.67 (16/24) | 66.67 (16/24) | 66.67 (16/24) | 45.83 (11/24) | 45.83 (11/24) | 4.17 (1/24) |
| Badger (*Meles meles*) | 17 | 29.41 (5/17) | 47.06 (8/17) | 47.06 (8/17) | 47.06 (8/17) | 35.29 (6/17) | 35.29 (6/17) | 0 (0/17) |
| Yellow throated marten (*Martes flavigula*) | 16 | 0 (0/16) | 50 (8/16) | 50 (8/16) | 50 (8/16) | 6.25 (2/16) | 6.25 (2/16) | 0 (0/16) |
| Korean hare (*Lepus coreanus*) | 14 | 0 (0/14) | 21.43 (3/14) | 21.43 (3/14) | 21.43 (3/14) | 0 (0/14) | 0 (0/14) | 0 (0/14) |
| Siberian weasel (*Mustela sibirica*) | 12 | 0 (0/12) | 16.67 (2/12) | 16.67 (2/12) | 16.67 (2/12) | 8.33 (1/12) | 8.33 (1/12) | 0 (0/12 |
| Siberian flying squirrel (*Pteromys volans*) | 11 | 0 (0/11) | 36.36 (4/11) | 36.36 (4/11) | 36.36 (4/11) | 18.18 (2/11) | 18.18 (2/11) | 0 (0/11) |
| Long tailed goral (*Naemorhedus caudatus*) | 4 | 0 (0/4) | 75 (3/4) | 75 (3/4) | 75 (3/4) | 0 (0/4) | 0 (0/4) | 0 (0/4) |
| Least weasel (*Mustella nivalis*) | 3 | 0 (0/3) | 33.33 (1/3) | 33.33 (1/3) | 33.33 (1/3) | 33.33 (1/3) | 33.33 (1/3) | 0 (0/3) |
| | 699 | 3.004 (21/699) | 45.63 (319/699) | 45.63 (319/699) | 45.63 (319/699) | 13.305 (93/699) | 13.305 (93/699) | 0.72 (5/699) |

[C] EMB" = *Escherichia coli* -selective eosin methylene blue and "CT-SMAC" = *Escherichia coli*-selective cefixime tellurite sorbitol MacConkey agar media

[d] "*stx-1+ stx-2* PCR" = *Escherichia coli* was detected by both primers of the Shiga toxin genes.

The numbers in parentheses indicate the percentage with number of the pathogen-detected samples divide by tested samples in close bracket.

[e] TSI = Triple Sugar Iron

"*" = significant result ($P < 0.05$) (exact value of $P = 0.042$, 95% CI (OR = 14.41; 1.10–190.99) via binary logistic regression analysis.

## Culture and molecular detection of commensal *E. coli* and STEC

Fecal cultures from 319 of the 699 samples formed colonies on *E. coli*-selective EMB agar and TSI media (**Tables 1 and 2**). To determine whether the colonies on EMB medium belonged to *E. coli*, PCR amplification of the bacterial 16S rRNA gene was performed. We randomly chose one colony at random from each of the 319 EMB agar plates, and after PCR amplification with the HVR primer set, we found target bands for the 16S rRNA gene in all the colonies (**Table 1**). Of the 319 fecal samples, 93 cultures that formed colonies on *E. coli*-selective EMB agar medium also formed colonies on STEC-selective CT-SMAC agar medium (**Table 2**).

To determine whether the colonies on CT-SMAC media belonged to STEC, PCR amplification of Shiga toxin genes (*stx1* and *stx2*) was performed. A single colony was randomly selected from each of the 93 CT-SMAC-positive agar plates (**Tables 1 and S3**). The *stx1* primer set produced a target band for the *stx1* gene in all 93 colonies, whereas the *stx2* primer set produced target bands for the Shiga toxin gene (*stx2*) and both genes (*stx1+stx2*) in five single colonies (**Table 1**).

## Prevalence pattern of *E. coli* and STEC in wildlife

*E. coli* was found in 45.63% (319/699) of the total samples, and two types of STEC with Shiga toxin genes, *stx1* and *stx1 + stx2*, were found in 13.31% (93/699) and 0.72% (5/699) of the samples, respectively. Further, STECs containing Shiga toxin genes *stx1+stx2* were found in the feces of water deer (n = 1), mouse (n = 2), wild boar (n = 1), and roe deer (n = 1) (Table 2).

With the exception of bird samples, *E. coli* was found in 44.29% (264/596) of mammalian samples, and two types of STEC with Shiga toxin genes, *stx1* and *stx1+stx2*, were found in 2.18% (13/596) and 0.84% (5/596) of the samples, respectively. Of the 103 bird samples, *E. coli* was detected in 53.40% (55/103) and STEC with the Shiga toxin gene *stx1* in 12.62% (13/103) of the samples. No STEC was found to carry Shiga toxin genes, *stx1+stx2* (Table 2).

*E. coli* was recorded in a high percentage of the fecal samples of the 13 species, including the long-tailed goral (75%), leopard cat (69.39%), and roe deer (66.67%). STEC with the Shiga toxin gene *stx1* was observed at the highest rate in the fecal samples of roe deer (45.83%), followed by badgers (35.29%) and weasels (33.33%). In addition, STEC with Shiga toxin genes *stx1+stx2* was found at the highest rate in fecal samples of roe deer (4.17%), followed by mice (3.63%), and wild boars (2.5%) (Table 2). Water deer accounted for the highest number of collected excrement samples (302). *E. coli* was detected in 35.43% (107/302) whereas STEC with the Shiga toxin gene *stx1* in 8.94% (27 samples), and STEC with the Shiga toxin genes *stx1 +stx2* in 0.33% (one sample) of the samples (Table 2).

In the case of *E. coli*, only bird fecal samples indicated a statistically (binary logistic regression analysis) significant result, which had an estimated odds of detection 14 times higher than that of the least weasel (OR = 14.41, 95% CI 1.10–190.99), and leopard cats had the second highest estimated odds of detection (OR = 9.258, 95% CI 0.73–117.86). In case of STEC, no significant result was observed, but roe dear had the highest estimated odds of detection (OR = 2.24, 95% CI 0.17–29.21) and significance level was at *p* = 0.54. Similarly, no significant results were observed for *Salmonella* (Table 2)

## Culture and molecular detection of *Salmonella*

Three *Salmonella*-selective media (SS, HE, and XLT4) were used to detect *Salmonella* species in fecal samples of 699 wild animals and birds. Fecal cultures from 21 samples formed *Salmonella*-positive colonies on SS, HE, and XLT4 agar media. Further, PCR amplification of *invA* and *iroB* was performed to confirm whether the colonies on SS, HE, and XLT4 media belonged to the genus *Salmonella*. A single colony was randomly selected from each positive-selection medium (Table 3). As a result, 27 colonies (21 lab-isolated and 6 reference-isolates, NCCP) were selected for the amplification of two genes (invasion gene, *invA*, and fur-regulated gene, *iroB*). All 27 colonies produced target bands following PCR amplification using *invA* and *iroB* gene primer sets (Table 3 and S5A and S5B Fig).

## Prevalence pattern of *Salmonella* in wildlife

*Salmonella* was detected in 21 (3%) of the 699 fecal samples from four of the 13 species (Table 2). *Salmonella* was found in the highest proportion in badgers (29.42%), followed by leopard cats (7.14%), magpie (6.8%), and wild boar (5%) whereas *Salmonella* was not detected in the other nine mammalian species, including water deer, from which the most feces were collected (Table 2).

**Table 3. Detection of 21 *Salmonella* isolates from 21 fecal samples.**

| No. Indiv.[*] | Animal name | Scientific Name | Fecal ID | Colony ID | Accession No | Size in bp (*mrcB* gene sequnce) | Cov.[&] | Ident.[$] | Collection[#] locality |
|---|---|---|---|---|---|---|---|---|---|
| 1 | Leopard cat | *P. bengalensis* | CaPrBe_3 | WC_A3 | OM793284 | 732 | 100 | 99.59 | BUS |
| 2 | Leopard cat | *P. bengalensis* | CaPrBe_5 | WC-A1-5-1 | OM793286 | 705 | 100 | 100 | BUS |
| 3 | Leopard cat | *P. bengalensis* | CaPrBe_10 | WC-A1-5-P | OM793287 | 723 | 100 | 100 | BUS |
| 4 | Leopard cat | *P. bengalensis* | CaPrBe_19 | WC-B4-5-2 | OM793290 | 729 | 100 | 99.73 | BUS |
| 5 | Leopard cat | *P. bengalensis* | CaPrBe_26 | WC-B4-8-2 | OM793289 | 750 | 100 | 99.73 | BUS |
| 6 | Leopard cat | *P. bengalensis* | CaPrBe_27 | WCB4_4 | OM793302 | 741 | 100 | 100 | BUS |
| 7 | Leopard cat | *P. bengalensis* | CaPrBe_28 | WC-B4-5-1 | OM793291 | 738 | 100 | 99.19 | BUS |
| 8 | Leopard cat | *P. bengalensis* | CaPrBe_31 | WC-B4-8-1 | OM793292 | 750 | 100 | 99.73 | BUS |
| 9 | Leopard cat | *P. bengalensis* | CaPrBe_39 | WC_B4(16) | OM793288 | 753 | 100 | 99.73 | BUS |
| 10 | Wild boar | *Sos. scrofa* | SuSuSc_1 | WB_235 | OM793285 | 735 | 100 | 100 | BUS |
| 11 | Wild boar | *S. scrofa* | SuSuSc_2 | WB_241 | OM793301 | 738 | 100 | 100 | HMS |
| 12 | Badger | *M. meles* | CaMeMe_1 | Badger-derong-I | OM793295 | 742 | 99 | 99.73 | CCG |
| 13 | Badger | *M. meles* | CaMeMe_3 | Raccon-BD-VI | OM793296 | 744 | 100 | 99.73 | TBG |
| 14 | Badger | *M. meles* | CaMeMe_4 | Raccon/BD-I | OM793299 | 735 | 100 | 100 | BHG |
| 15 | Badger | *M. meles* | CaMeMe_9 | badger-derong-100 | OM793303 | 743 | 100 | 100 | CCG |
| 16 | Magpie | *Pica cericea* | ABF_42 | BF-3-KNU-I | OM793293 | 741 | 100 | 99.73 | SKG |
| 17 | Magpie | *P. cericea* | ABF_43 | BF_3_KNU (ii) | OM793300 | 744 | 100 | 100 | SKG |
| 18 | Magpie | *P. cericea* | ABF_47 | BF-11-041016 | OM793297 | 741 | 100 | 99.6 | SKG |
| 19 | Magpie | *P. cericea* | ABF_48 | BF-1-ch-041016-i | OM793298 | 742 | 99 | 99.73 | SKG |
| 20 | Magpie | *P. cericea* | ABF_58 | BF-11-ch-041018 | OM793304 | 744 | 100 | 99.87 | CCG |
| 21 | Magpie | *P. cericea* | ABF_77 | BF-3-KNU-17 | OM793294 | 747 | 100 | 99.87 | CCG |

[#]The collection localities abbreviations are as follows: Bus: Busan; CCG: Chuncheon, Gangwon-do; TBG: Taebaek, Gangwon-do; BHG: Gyong-buk- Bonghwa; SKG: Sokcho/ Seorak, Gangwon-do; HMS: Hanam-si-Seoul

[*]indiv = Individual

[&]Cov. = Coverage of NCBI BLAST searching of a specific gene

[$]Ident. = Identity of NCBI BLAST searching of a specific gene.

## Phylogenetic analysis of *E. coli* and *Salmonella*

The STEC isolate sequences were amplified with *ileS* sequences selected from 93 *stx1* positive animal fecal samples for phylogenetic analysis, which were used for NJ and ML analysis. A total of 93 *E. coli* isolates belonged to *stx1*, including three major lineages (groups 1–3); the *stx1*+ *stx2*-positive group is marked by a shaded pastel pink color (Fig 2).

The MJN indicated haplotypes (n = 64, denoted as Hap-1–Hap-64) in animal hosts (Fig 3) and places (S7 Fig) to the respective MJN clade and phylogenetic analysis, and the gene sequences were collected from wild animals. A total of 64 haplotypes were identified in all the South Korean *E. coli* isolates (n = 93) in the *ileS* sequence (774 bp, number of polymorphic sites = 64) (S3 Table). In addition, the South Korean isolates were well grouped with *stx1* +*stx2*-positive *E. coli*, including water deer (n = 1, H-59), roe deer (n = 1, H-12), rodents (n = 2, H-7 and H-12), and wild boar (n = 1, H-12), which are marked by green, yellow green, and yellow color, respectively (Fig 3 and S3 Table).

The 21 *Salmonella* spp-positive samples (n = 21, isolates collected from each of the positive *Salmonella* fecal samples) were sequenced with partial *mrcB* sequences. We also sequenced six South Korean reference *Salmonella enterica* serovar strains, including *S.* Abony (accession no. OM793305), *S.* Typhimurium (accession no. OM793306), *S.* Enteritidis (accession no.

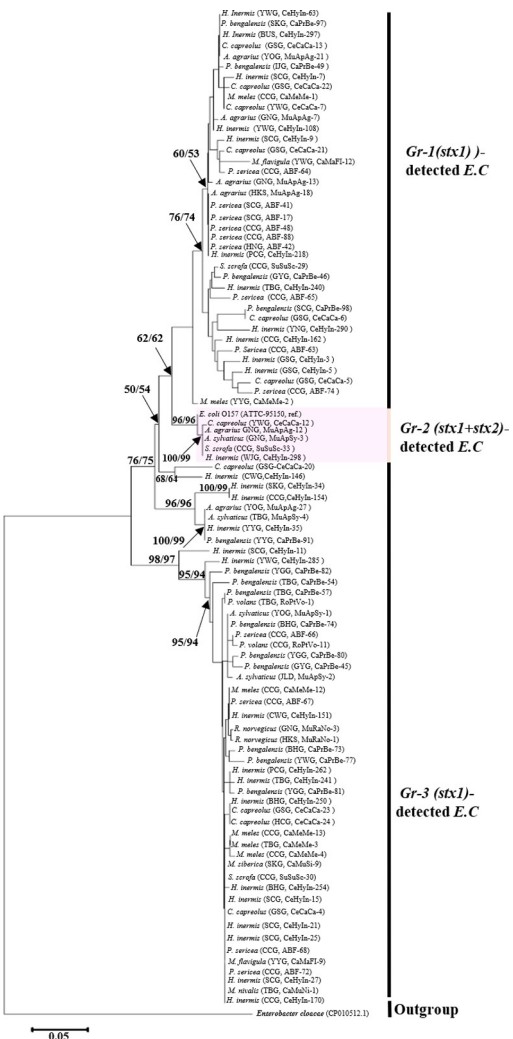

**Fig 2. Phylogenetic tree of isoleucine—tRNA ligase gene (*ileS*, partial gene sequence, 774 bp) of Shiga-toxin producing *Escherichia coli* (STEC).** The isolated *E. coli* strains were collected from Korean wild animal feces and they were separated into three major lineages based on *ileS* sequences [68]. The (*stx1*+ *stx2*)-positive group marked by shaded light orange color. The bootstrap values are provided above branches of tree, and the given bootstrap numbers are indicated as neighbor-joining (NJ) and maximum likelihood (ML) methods. (*stx1*+ *stx2*)-detected group marked by shaded light orange color. Scientific names of animals and a detailed sample information (sample locality code, sample number code) are provided in S1 Table.

OM793307), and *S.* Agona (accession no. OM793308), *S. enterica* subsp. *enterica* (accession no. OM793309), and *S.* Typhi (accession no. OM793310). Of the 27 *Salmonella* isolates from wild animal fecal samples (n = 21, positive number of wild animal fecal samples), we used 21 isolates for gene network and phylogenetic analysis and six isolates as reference *Salmonella* sequences. The resulting phylogenetic tree indicates the clusters that belong to two main *Salmonella* subspecies, *S. enterica* subsp. enterica (the reference *Salmonella* group of *S. enterica* subsp. *enterica* typhimurium [*S. typhimurium*], *S. enteritidis*, *S. anatum*, *S. agona*, and *S. enterica* groups), and another being *S. enterica* subsp. *enterica* diarizonae (Figs 4 and 5). A total isolate (n = 54) of the *mrcB* (amplicon length = 763 bp) sequences of *Salmonella* spp., including the NCBI acquired gene sequences (*n* = 27), laboratory-isolated *Salmonella* gene sequences (*n* = 21), and reference South Korean culture collection *Salmonella* serovar strains (*n* = 6) were

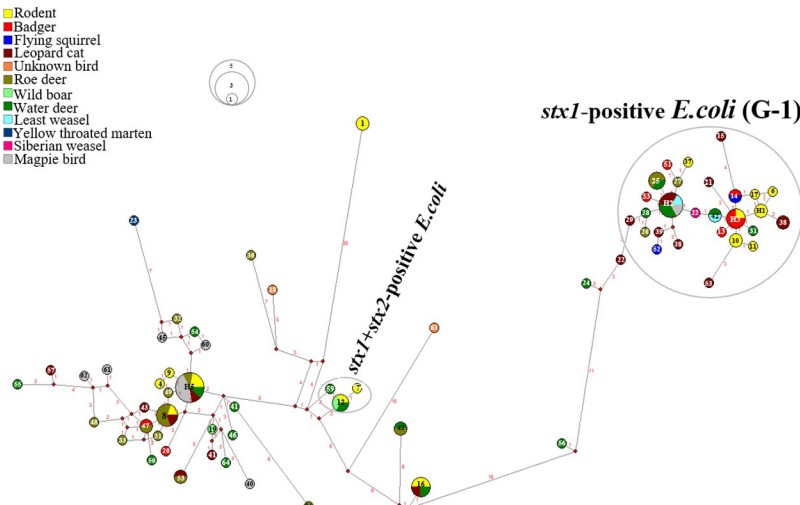

**Fig 3. The median-joining network (MJN) indicating the relationship of *stx1* and *stx1+stx2*-positive *Escherichia coli* with isoleucine—tRNA ligase gene (*ileS*) sequence.** The bacterial-borne zoonotic pathogen Shiga toxin genes detected in *E. coli isolated* from the feces of wild mammals and birds in South Korea.

used in the phylogenetic analysis. Fig 4 depicts two species of *Salmonella*: *S. enterica* and *S. bongori*. Polymorphic sites (*n* = 43) of *Salmonella* were observed in the alignment of laboratory-isolated (*n* = 21 isolates) and reference *Salmonella* sequences (n = 6 isolates), followed by the South Korean isolates. The haplotype diversity was 0.9231. The haplotype information is provided in S3 Table. Higher bootstrap values seem to be more accurate in the phylogenetic tree obtained using MEGA software. The partial *mrcB* sequences of ten haplotypes found in South Korean *Salmonella* isolates (n = 21 *Salmonella* strains) are marked by a star in Fig 1, which can be accessed via GenBank with accession numbers OM793284–OM793304 [67]. The MJN indicated the haplotypes (n = 21, denoted as Hap-1–Hap-21) in animal hosts (Fig 5A) and places (Fig 5B) in the respective MJN clade, and phylogenetic analysis and gene sequences were collected from South Korean wild animals (n = 21), NCBI acquired (n = 27), and reference *Salmonella* (n = 6).

## Geographical and seasonal prevalence pattern of *E. coli* and *Salmonella*

Twenty-one *Salmonella*-positive samples (from 699 samples) were collected from six areas, including three in Gangwon Province (Sokcho, SKG; Chuncheon, CCG; and Taebaek, TBG) and three in areas outside Gangwon Province (Hanam-si, Seoul, HKS; Bonghwa, Gyong-buk-do, BHG; and Busan, BUS) (Table 3). *Salmonella* was detected in the highest proportion in Busan City (25.64%) (10 *Salmonella* positive samples out of 39 tested-fecal samples). Within Gangwon province, *Salmonella* was detected at a higher level in Samcheok (5.45%) than in the other two areas (4.44% in Chuncheon and 2.27% in Taebaek) (Table 1 and S1 Fig).

Notably, STEC with the Shiga toxin gene *stx1* was detected in 21 of 25 survey areas, whereas STEC with Shiga toxin genes *stx1+stx2* was detected in only four areas, including three in Gangwon Province (Yeongwol, YWG; Wonju, WJG; and Chuncheon, CCG), and one in another province (Geochang, GNG) (Table 1 and S1 and S6 Figs). Also, STEC with Shiga toxin gene *stx1* was detected at the highest ratio (43.48%) in Goseong, Gangwon province, followed by Sokcho (38.46%) and Yangyang (35.29%) in Gangwon province (Table 1 and S1 and S6 Figs). Shiga toxin genes *stx1+stx2* were detected at the highest ratio (20.00%) in Wonju, followed by Geochang (8.70%) (Table 1).

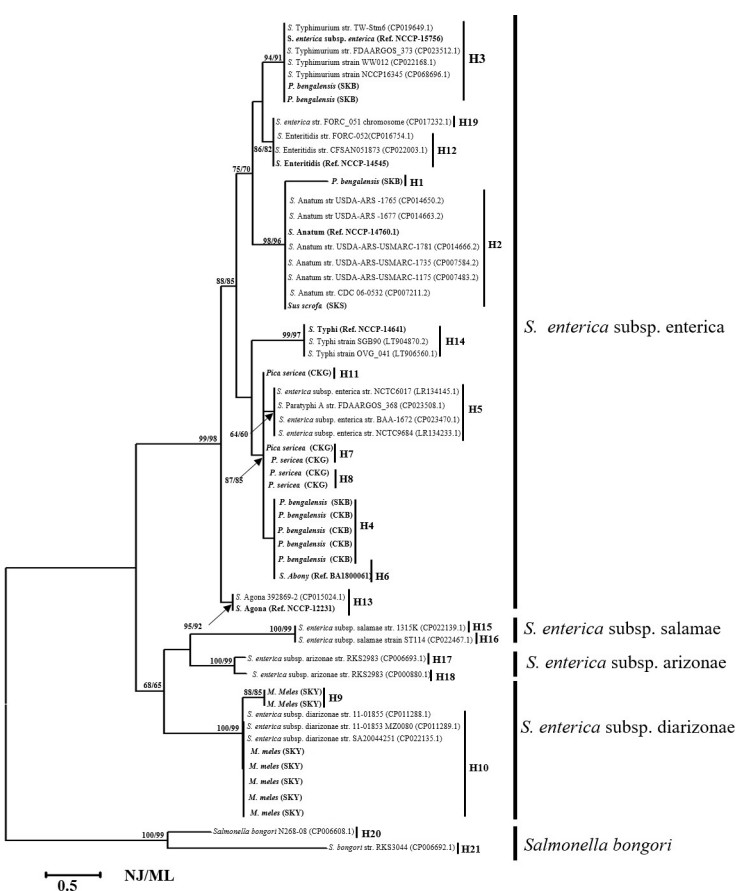

**Fig 4. Phylogenetic analysis of *Salmonella* (South Korean- and worldwide-isolated sequence) and their haplotype number in parenthesis with each group of *Salmonella*.**

The effect of seasons (summer, spring, and fall), sample collection areas (five different directions in 25 counties), and animal samples (approximately 700 wild animals and single bird feces) seem to have an effect on the prevalence results. Therefore, regression analysis was conducted on all sample collections or survey areas, with each pathogenic agent as outcome variables. However, no statistically significant difference ($P = 0.05$) was found among the different regions of different collection sites in South Korea with respect to the pathogenic group of bacteria (*Salmonella*, *E. coli*, *stx-1*-positive STEC, and *stx1* + *stx2*-positive STEC).

With the exception of *Salmonella*, of the 151 samples collected in spring, the percentage of positive samples was the highest in spring followed by summer and fall, indicating a seasonal effect on the detected pathogens (Fig 6 and S3 Table). When results were checked by each season, prevalence of *E. coli* was significant in spring (56.95%; 95% CI = 6.13–24.07) and summer (46.50%; 95% CI = 1.804–5.454), while the prevalence of *stx1*-positive STEC was significant in spring (17.22%; 95% CI = 1.419–8.388) (Fig 6). *E. coli* detection was six times higher (OR = 12.15) and STEC detection was three- and a half times (OR = 3.45) higher compared with that of autumn.

During summer (June to August), the highest percentage (3.50%) and number (14 of 400) of positive *Salmonella* samples were detected (Fig 6 and S2 Table). On the other hand, October 2016 indicated the highest percentage of *Salmonella* samples (3 of 23; 13.04%), while June 2016 had the highest number of *Salmonella*-positive samples (9 of 110; 8.18%) (S3 Table). When

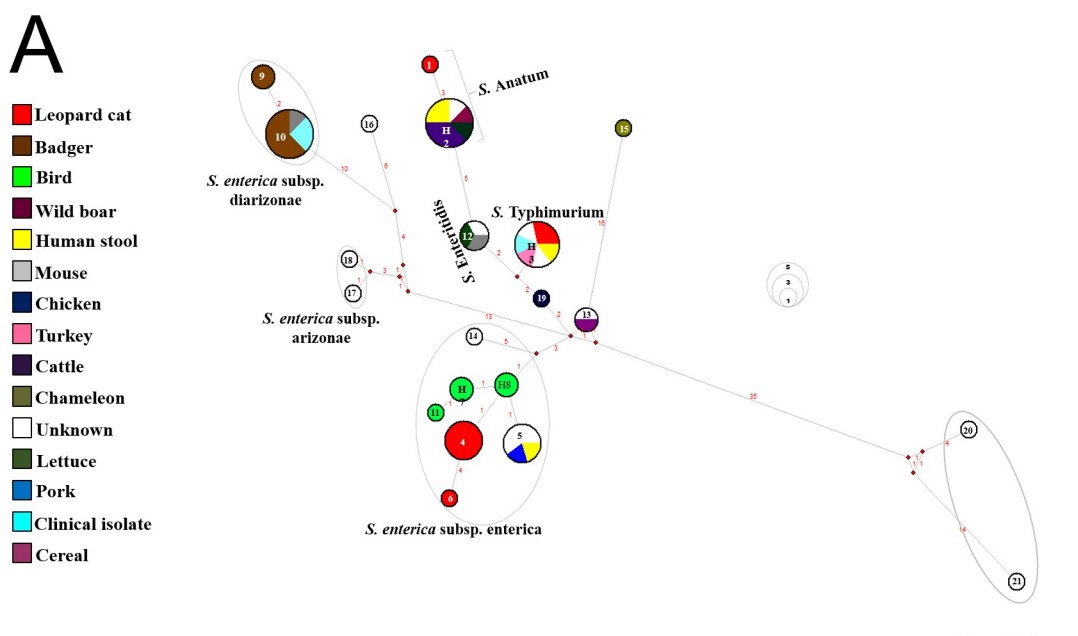

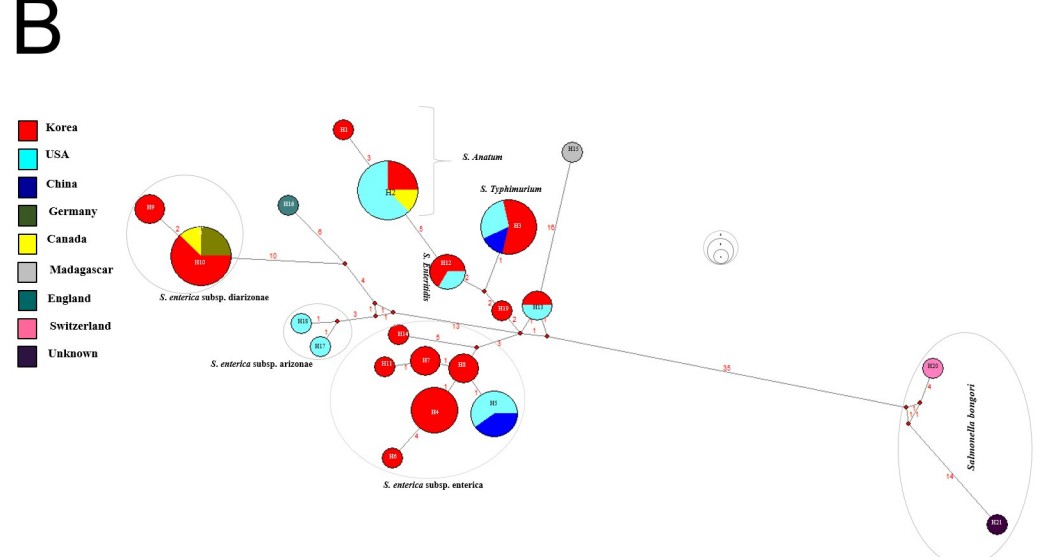

**Fig 5.** The median-joining network (MJN) indicating the relationship of penicillin-binding protein (*mrcB*) gene sequence of *Salmonella* with the hosts (A) and localities (B) of animals.

*Salmonella* prevalence was checked by season, no significant difference was observed ($P>0.05$) among the three seasons (Fig 6 and S3 Table).

In the case of Shiga toxin gene, *stx1* detection in July (OR = 8.311, 95% CI = 2.48–27.83) and October (OR = 13.2, 95% CI = 3.22–54.13) showed statistically significant ($P<0.001$) results (S3 Table). Similarly, April and May indicated significant results ($P<0.001$). During the spring season, however, strains containing the highest percentage of total *E. coli* (56.95%), and Shiga toxin genes *stx1* (17.22%), and *stx1*+stx2 (1.33%) were detected (Fig 1).

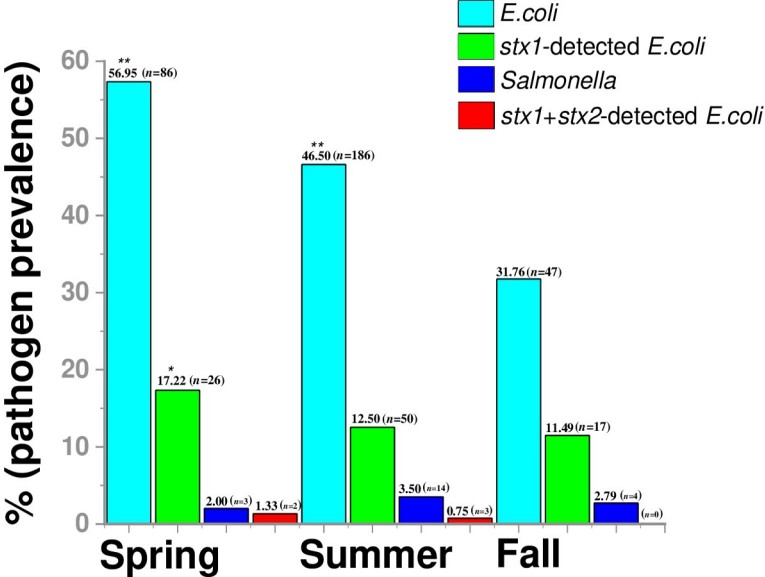

**Fig 6. Seasonal prevalence of commensal *Escherichia coli*, *stx1* and *stx2*-positive *E. coli*, non-typhoidal *Salmonella*
(NTS).** Samples were not collected in winter, majority of the samples were collected in the three seasons, three months
each of spring (Mar–May), summer (Jun–Aug), and autumn (Sep–Nov). ** indicates $P<0.001$, *indicates the $P<0.005$.
The detailed sample information, ages is provided in S3 Table. The color bar aqua, green, indigo, and red indicates
*Escherichia coli* (aqua), *stx1*-positive *E. coli* (green), *stx1* and *stx2*-positive *E. coli* (indigo), *Salmonella* (red),
respectively.

## Discussion

*Salmonella* and *E. coli* are commonly found in the gastrointestinal tract of wild animals, rep-
tiles, and many bird species and are the most common contaminants in agricultural and ani-
mal products. The most pathogenic serotype of *E. coli* is O157:H7, which produces Shiga
toxins (stx1 and stx2) that cause severe hemolytic-uremic syndrome (HUS) in humans. In this
study, two Shiga toxin genes (*stx1* and *stx2*) were identified in the feces of wild animals
(Table 1). Further, *stx1* was detected in animal feces (Table 2), at a higher number than that of
*stx2*, which has been established in many previous studies [74, 75]. The Korean Centers for
Disease Control and Prevention (KCDC) conducted a survey on water- and food-borne dis-
ease outbreaks from 2007 to 2009. In this survey, a total of 1026 outbreaks were observed,
resulting in infections in 25,310 patients, with 913 reported outbreaks (89.0%). Norovirus was
the most common causative agent (16.5%), followed by pathogenic *E. coli* (13.9%), and *Salmo-
nella* spp. (7.7%) [76]. Furthermore, only two O157 serotypes were identified in the fecal sam-
ples of South Korean cattle in a study conducted between September 2010 and July 2011,
whereas other serotypes were detected in cattle feces [77]. In this study, only five *stx2*-produc-
ing *E. coli* strains were detected (Fig 2). Several studies have found that wild boar [45, 78],
domestic pigs [79], wild deer [80], domestic and wild cattle [45], sheep, and goats [81] are the
most significant potential sources of *E. coli* O157:H7, which can easily spread to agricultural
leafy green production areas [45]. Deer feces was found in outbreaks associated with agricul-
tural produce in the western United States [52], those caused by contamination of fresh straw-
berries in Oregon [52], those caused in Sweden due to contaminated lettuce [82], and those
linked to unpasteurized apple juice [83]. A survey of deer feces has revealed low levels ($< 1\%$)
of *E. coli* O157 [84]; similarly, in this study, low levels of *E. coli* (*stx1* and *stx2*) were detected in
the feces of water deer (0.33%, n = 302) and roe deer (4.17%, n = 24) in South Korea (Table 2).

Moreover, owing to their unique ability to travel long distances, free-roaming wild birds can disseminate *Salmonella* and STEC to various environments [24, 85–88]. Wild animal environmental feces can contaminate agricultural produce [89] and infect farmers, owing to the long-term survival of STEC and *Salmonella* in feces. In a study [90], the overall *Salmonella* pathogen prevalence in all analyzed samples was 1.4%, and in another study [84], *Salmonella* prevalence in free-ranging deer feces was 1%, indicating a detection trend similar to that found in our samples (Table 3). Several studies have found that many competing microbiotas are present in a sample, making it difficult to identify STEC during the culture process [20, 91, 92]. Moreover, owing to a variety of sub-cultivation processes, STEC strains may easily lose *stx1* and *stx2* [92]. This could be a reason why we found fewer Shiga toxin genes in *E. coli*.

In a recent study, outbreaks between 2015 and 2019 revealed that the causative norovirus was the most common pathogenic agent, accounting for 21.6% of cases. Aside from norovirus, 11.9% were others, including 6.3% *E. coli*, and 3.5% *Salmonella* spp. From May to October, the majority (80.4%) of outbreaks involved bacterial pathogens, such as pathogenic *E. coli*. Outbreaks occurred most intense from May to September—9.5%, 9.2%, 9.3%, 9.8%, and 10.2%, respectively, accounting for 48% of the total annual outbreaks [11].

Previous studies have identified *Salmonella* spp. from wild animals, including badgers, wild boars, and coyotes [48, 93, 94]. In this study, wild leopard cats and badgers had the highest frequency of *Salmonella* spp (Table 2). This result supports the hypothesis of previous studies (78) that wild carnivores can be natural reservoirs of *Salmonella* spp. Furthermore, pathogen detection varies according to geographical region and other factors such as differences in environmental pressure, moisture, temperature, sunlight (ultraviolet), nutrients, and possibly unknown factors that enhance pathogen incensement. All these factors may be influenced by the status of good agricultural practices (GAPs) and implementation of national measures in accordance with national or international guidelines, as well as the support available to farmers involved in agricultural practices [95, 96]. Notably, GAPs were first introduced in Korea in 2003; they help to improve food safety [97] by reducing the number of pathogenic organisms that can survive in agricultural environments and foods. The goal of GAPs in Korea is to ensure the safety of agri-food, increase the trust of domestic consumers, increase the competitiveness of Korean agri-food in the international market, and protect the agricultural environment from pathogenic microbes, such as *Salmonella* and *E. coli*. As previously stated, most *Salmonella* strains were isolated in June [98]. The highest number of *Salmonella* spp. was detected in June (S3 Table). Moreover, a study conducted by Flores-Monter et al. in 2021 revealed a relationship between temperature, humidity, and seasonal variation in the persistence of *Salmonella* spp. in the environment [99]. In this study, seasonal variations in the presence of *Salmonella* and *E. coli* were observed during season-dependent environmental parameters (Fig 6). The number of *Salmonella*- and *E. coli*- induced outbreaks has recently increased in Korea [37, 39]. The majority (80.4%) of outbreaks involving bacterial pathogens such as pathogenic *E. coli*, occurred between May and October. Between May and September, outbreaks were most common (9.5%, 9.2%, 9.3%, 9.8%, and 10.2%, respectively), accounting for 48% of all annual outbreaks [11]. Another survey of diarrheagenic bacteria in the stools of patients (2444/21180, 15% bacteria causing diarrheal detection, pathogenic *E coli* being one of them) found that the highest isolation rate was observed between June and September in Korea [100], which was followed by our current study.

Furthermore, although different enrichment methods for *Salmonella* and pathogenic *E. coli* detection are frequently used, the same primary enrichment cultures were reported for the isolation of both pathogens [15]. In our study, the same primary enrichment culture method was used to detect *Salmonella* and *E. coli*. In this study, we found that black-colored colonies on SS agar did not produce a black color on HE and XLT4 agar. As a result, the HE and XLT4 agar

media were used for the consistent and reliable identification of selective culture media for *Salmonella* identification across multiple samples.

Phylogenetic and gene network analyses were performed to determine the grouping of *Salmonella* and *E. coli*. The *Salmonella* isolated from wild animal feces in South Korea were well cladded into two major subspecies, including *S. enterica* subsp. *enterica* (haplogroup, H-3, -4, -7, -8, -11, and -12) and *S. enterica* subsp. diarizonae (H-9 and -10). Similarly, 93 detected *E. coli* were grouped into two major groups: *stx1*-positive *E. coli* and (*stx1*+ *stx2*)-positive *E. coli*. Further *stx1*-positive *E. coli* strains were divided into two groups (*stx1* positive group-1 and -2) based on the sequence matched with those in GenBank (https://www.ncbi.nlm.nih.gov/). Similarly, a previous study indicated that SNP-based phylogenetically grouped *Salmonella* strains and indicated their the genetic diversity [101]. The level of gene or haplotype diversity among South Korean *Salmonella* strains was found, haplotype diversity = 0.9231, indicating the genetic variation of Korean *Salmonella* strains. The variable sites were polymorphic n = 46, and within the 707 bp aligned gene sequences, and 13 haplotypes were observed among the 27 *Salmonella* isolates. Furthermore, the genetic diversity among *Salmonella* isolates (n = 54 sequences, 693 bp aligned sequences, and haplotype 21) was observed. Six haplotypes (H-1, -4, -6, -7, -8, -11) from South Korea and one shared haplotype (H-5) from China were grouped into *S. enterica* subsp. enterica. Similarly, Korean and Chinese *Salmonella* isolates were well classified into the following groups: *S. enterica* subsp. diarizonae, *S. anatum*, *S. enteritidis*, and *S. typhimurium* (Fig 5).

A limitation of this study is the molecular typing of a specific *E. coli* phylogroup. Without serotyping, we could not perform the other non-O157 STEC typing. We observed false-positive and false-negative results for the cultural identification of *Salmonella* and *E. coli*, respectively. *E. coli* was detected in approximately 67% of cultures with traditional culture methods, while *E. coli* was detected in approximately 45.63% of cultures with molecular analysis (Table 2). To confirm the accurate prevalence of *E. coli*, and phylogroup of *Salmonella* and *E. coli*, further studies with a large number of fresh fecal and diverse wild animal samples and wide-scale whole-genome sequencing for surveillance and outbreak tracing are needed.

In conclusion, *E. coli* and *Salmonella* are commonly found in the lower intestines of mammals, but these organisms can also live outside the body and persist in the environment for several months to more than a year. Based on our observations, wildlife samples are a source of fecal-borne pathogens in various regions of South Korea. Irrigation water and rain water may be the cause of fecal-borne pathogens dissemination in Korea [102, 103]. Monitoring *Salmonella* in this region could aid in determining the temporal and spatial factors that influence *Salmonella* incidence, as well as in source tracking, higher resolution of different serovars, and comparisons with similar data from other agricultural regions in South Korea. Identifying *Salmonella* and *E. coli* O157 in agricultural production regions and other important production regions in South Korea is critical for improving wildlife management and public health, particularly the health of farmers. Identifying these reservoirs and understanding the dynamics of human infections are essential for developing control strategies to reduce *Salmonella* and *E. coli* infections in the human population. The research data facilitates a better understanding of the potential risk of wildlife contamination, which could be used to improve public health (associated with food-borne diseases) prevention strategies.

## Supporting information

**S1 Fig. Maps showing the sample collection information and prevalence pattern of commensal *Escherichia coli*, *stx1* and *stx2*-detected *E. coli*, *Salmonella* from fecal at Gangwon province in South Korea.** "*N*" = A total collection samples in each region and color-coded

round shape ratio indicate *Escherichia coli* (blue), *stx1*-detected *E. coli* (orange), *stx1* and *stx2*-detected *E. coli* (red-orange), *Salmonella* (red), respectively.
(TIF)

**S2 Fig.** Shows the normal structure and color of collected feces samples A) water dear (H. inermis) feces in a round shape or heart shape with convex wall B) white urea-containing liquid with an amorphous shape of Pica sericea feces C) leopard cat fecal tubular shape of leopard cat (Prionailurus bengalensis) D) wild boar (Sus scrofa) feces E). wild boar (Sus scrofa) fecal with corn seed F) badger (Meles meles) feces with tubular-shape G) yellow throated marten (Martes flavigulla) H) Korean hare (Lepus coreanus) I). flying squirrel (Pteromus volans) and J). least weasel (Mustella nivalis).
(TIF)

**S3 Fig.** A–D. The sample collection sites, A) Sherman traps were kept in an orchard garden, Gangwon-do, B) Survey area of the Chuncheon agricultural plots, Gangwon-do, C) Survey area of the nearby streams of water where samples are collected, and D) Survey area of Bukhansan, a forested region where samples are collected.
(TIF)

**S4 Fig. Representative survey area and camera trapping at Odaesan National Park, Pyeongchang, Gangwon-do, Korea.** 1–10 represent camera trapping, and A and B indicate survey area location.
(TIF)

**S5 Fig.** (A, B). PCR amplification for Salmonella detection with invasion gene (invA) primer set PCR bands were observed in all 27 single colonies. The 27 isolated strains (n = 21, the isolated strains from wild animal fecal sources and the most widely prevalent reference Salmonella serovars, n = 6) were used. PCR band 'M' indicates DNA 100 bp marker. The target band of the amplified invA primer was 398 bp. The gel lane numbers are as follows: from No.1 to 6 reference serovars and Salmonella-positive isolated strains were detected from wild animal fecal between lanes No.7 and No.27 (B) PCR amplification for Salmonella detection with the iron chelating (iroB) gene primer set. PCR bands were observed in all 27 single colonies. PCR band 'M' indicates DNA 100 bp marker. The target band of the amplified iroB primer was 606 bp. The Lane from No.1 to 6 reference serovars and Salmonella-positive isolated strains were detected from wild animal fecal between lanes from No.7 and No.27. The symbol "–"PCR mixture without genomic DNA of samples was used as a negative control. Each sample is indicated by a host individual ID with bacterial colony ID in parentheses. Lane from No.1 to 6 reference serovars [S. Abony (BA1800061), S. enteritidis (NCCP-14545), S. Agona (NCCP-12231), S. Typhimurium (NCCP-14760), S. Typhi (NCCP-14641), and S. Enterica (NCCP-15756)]; (Lane No. 7 = CaPrBe_3 (WC_A3); Lane No. 8 = CaPrBe_5 (WC-A1-5-1); Lane No. 9 = CaPrBe_10 (WC-A1-5-P); Lane No. 10 = CaPrBe_19 (WC-B4-5-2); Lane No. 11 = CaPrBe_26 (WC-B4-8-2); Lane No. 12 = CaPrBe_27 (WCB4_4); Lane No. 13 = CaPrBe_28 (WC-B4-5-1); Lane No. 14 = CaPrBe_31 (WC-B4-8-1); Lane No. 15 = CaPrBe_39 (WC_B4 (16); Lane No. 16 = SuSuSc_1 (WB_235); Lane No. 17 = SuSuSc_2 (WB_241); Lane No. 18 = CaMeMe_1 (Badger-derong-I); Lane No. 19 = CaMeMe_3 (Raccon-BD-VI); Lane No. 20 = CaMeMe_4 (Raccon/BD-I); Lane No. 21 = CaMeMe_9 (badger-derong-100); Lane No. 22 = ABF_42 (BF-3-KNU-I); Lane No. 23 = ABF_43 (BF_3_KNU(ii)); Lane No. 24 = ABF_47 (BF-11-041016); Lane No. 25 = ABF_48 (BF-1-ch-041016-i); Lane No. 26 = ABF_58 (BF-11-ch-041018); Lane No. 27 = ABF_77 (BF-3-KNU-17).
(TIF)

**S6 Fig. Schematic diagram of SNP-encompassing penicillin-binding protein gene (*mrcB*) primer (mrcB-9-Sbon-F/R) of a reference *Salmonella bongori* (NC-015761; 2523 bp) for amplification of target *Salmonella mrcB* gene sequences.** The first primer set of forward primer, Sbon-F (19 bp) was between 1362 and 1379, and reverse primer Sbon-R (21 bp) was between 2160 and 2142, respectively and the target band was approximately 793 bp. Red color "SNP-1878" indicates the bases either T/ C/G mutated/replaced with each other. The natural encompassing SNP position was marked by red color in the position of 1878 of the reference (NC-015761) *mrcB* gene sequence.
(TIF)

**S7 Fig. Gene-network analysis of isoleucine—tRNA ligase gene (*ileS*) gene sequences of Shiga toxin-producing *Escherichia coli*.** Distribution pattern (locality) of bacterial-borne zoonotic pathogen Shiga toxin genes (*stx1* and *stx1*+*stx2*)-detected *E. coli* from the feces of wild mammals and birds in South Korea.
(TIF)

**S1 Table. Geographic location and haplotype information of *stx-1*-detected *Escherichia coli* isolated from wildlife feces in South Korea.**
(DOCX)

**S2 Table. Information on Shiga toxin genes (*stx1*), and (*stx1*+*stx2*)-detected *Escherichia coli* positive samples.**
(DOCX)

**S3 Table. Monthly and seasonal sample information and prevalence of *Salmonella*, *Escherichia coli*, Shiga toxin genes (*stx1*), and (*stx1*+*stx2*)-detected *E. coli*.**
(DOCX)

## Acknowledgments

We thank the laboratory members of Wildlife Genomics at the Kangwon National University. In particular, we acknowledge the support of Kwang Bae Yoon, a senior researcher at the National Institute of Ecology, for sample collection.

## Author Contributions

**Conceptualization:** Lim Sangjin, Park Y. Chul.

**Data curation:** Rahman M. Mafizur.

**Formal analysis:** Rahman M. Mafizur, Lim Sangjin.

**Investigation:** Rahman M. Mafizur, Lim Sangjin.

**Project administration:** Lim Sangjin, Park Y. Chul.

**Resources:** Lim Sangjin.

**Software:** Rahman M. Mafizur.

**Supervision:** Park Y. Chul.

**Validation:** Rahman M. Mafizur.

**Visualization:** Rahman M. Mafizur.

**Writing – original draft:** Rahman M. Mafizur, Park Y. Chul.

Writing – review & editing: Park Y. Chul.

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
