## [Decision Letter · Decision Letter 0]

28 Sep 2022

PONE-D-22-23426Prevalence of Salmonella spp. and Escherichia coli in the feces of free-roaming wildlife throughout South KoreaPLOS ONE

Dear Dr. Yung Chul Park,

Thank you for submitting your manuscript to PLOS ONE. After careful consideration, we feel that it has merit but does not fully meet PLOS ONE’s publication criteria as it currently stands. Therefore, we invite you to submit a revised version of the manuscript that addresses the points raised during the review process.

ACADEMIC EDITOR:Please address all of the reviewers' comments.Take particular attention to  describing and explaining the association between wildlife, agricultural produce, and food poisoning of humans in South Korea. Describe also the risk factors that might impacted the contamination of produce by wildlife. 

We look forward to receiving your revised manuscript.

Kind regards,

Csaba Varga, DVM MSc PhD

Academic Editor

PLOS ONE

Journal Requirements:

3. As part of your revision, please complete and submit a copy of the Full ARRIVE 2.0 Guidelines checklist, a document that aims to improve experimental reporting and reproducibility of animal studies for purposes of post-publication data analysis and reproducibility: https://arriveguidelines.org/sites/arrive/files/documents/Author%20Checklist%20-%20Full.pdf Please include your completed checklist as a Supporting Information file. Note that if your paper is accepted for publication, this checklist will be published as part of your article.

4. In your Methods section, please provide additional information regarding the permits you obtained for the work. Please ensure you have included the full name of the authority that approved the field site access and, if no permits were required, a brief statement explaining why.

"This work was supported by “Cooperative Research Program for Agricultural Science & Technology Development (Project No. PJ0108592016), Rural Administration, Republic of Korea. We thank the laboratory members of Wildlife Genomics at Kangwon National University. In particular, we acknowledge the support of Kwang Bae Yoon, a senior researcher at the National Institute of Ecology, for collecting samples."

"Not applicable "

7. PLOS requires an ORCID iD for the corresponding author in Editorial Manager on papers submitted after December 6th, 2016. Please ensure that you have an ORCID iD and that it is validated in Editorial Manager. To do this, go to ‘Update my Information’ (in the upper left-hand corner of the main menu), and click on the Fetch/Validate link next to the ORCID field. This will take you to the ORCID site and allow you to create a new iD or authenticate a pre-existing iD in Editorial Manager. Please see the following video for instructions on linking an ORCID iD to your Editorial Manager account: https://www.youtube.com/watch?v=_xcclfuvtxQ.

8. We note that you have included the phrase “data not shown” in your manuscript. Unfortunately, this does not meet our data sharing requirements. PLOS does not permit references to inaccessible data. We require that authors provide all relevant data within the paper, Supporting Information files, or in an acceptable, public repository. Please add a citation to support this phrase or upload the data that corresponds with these findings to a stable repository (such as Figshare or Dryad) and provide and URLs, DOIs, or accession numbers that may be used to access these data. Or, if the data are not a core part of the research being presented in your study, we ask that you remove the phrase that refers to these data.

9. Your ethics statement should only appear in the Methods section of your manuscript. If your ethics statement is written in any section besides the Methods, please move it to the Methods section and delete it from any other section. Please ensure that your ethics statement is included in your manuscript, as the ethics statement entered into the online submission form will not be published alongside your manuscript. 

10. We note thatFigure 1 in your submission contain [map/satellite] images which may be copyrighted. All PLOS content is published under the Creative Commons Attribution License (CC BY 4.0), which means that the manuscript, images, and Supporting Information files will be freely available online, and any third party is permitted to access, download, copy, distribute, and use these materials in any way, even commercially, with proper attribution. For these reasons, we cannot publish previously copyrighted maps or satellite images created using proprietary data, such as Google software (Google Maps, Street View, and Earth). For more information, see our copyright guidelines: http://journals.plos.org/plosone/s/licenses-and-copyright.

Reviewers' comments:

Reviewer's Responses to Questions

**Comments to the Author**

1. Is the manuscript technically sound, and do the data support the conclusions?

Reviewer #1: Partly

Reviewer #2: Yes

Reviewer #3: Partly

2. Has the statistical analysis been performed appropriately and rigorously? 

Reviewer #1: No

Reviewer #2: Yes

Reviewer #3: No

3. Have the authors made all data underlying the findings in their manuscript fully available?

Reviewer #1: Yes

Reviewer #2: Yes

Reviewer #3: No

4. Is the manuscript presented in an intelligible fashion and written in standard English?

Reviewer #1: No

Reviewer #2: Yes

Reviewer #3: Yes

5. Review Comments to the Author

Reviewer #1: The manuscript did not thoroughly explain the association between wildlife, agricultural produce, and food poisoning in South Korea. The authors did not fully present the logic or reference whether the food poisoning in South Korea was caused by the consumption of agricultural produce which might be contaminated by wildlife feces. Statistical analysis was not explained comprehensively. This manuscript did not present in an intelligent fashion: more than ten sentences in the Introduction and Discussion sections require adding or changing references. The naming of the collection points was not consistent and a lot of typos and punctuation errors. It is recommended to use a professional copyediting service.

Line 50: References are not sufficient. The authors talked about wild animals but references 1-4 were about chickens, food animals, and humans.

Line 51: Add reference for humans.

Lines 51-53: The sentence is confusing. Is fecal-oral transmission the only transmission route? Salmonella can spread from animals to people and from people to people.

Lines 57-58: Is this for humans or is this include animals? Bai et al. (2016) reported non-O157 STEC.

Lines 59-60: Mogren et al (2018) discussed the hurdle approach to control pathogenic bacterial contamination of leafy green vegetables.

Line 62: High prevalence is not clear. It is suggested to elaborate on the prevalence and host ubiquity written by Kim et al. (2020)

Line 67: Add areas.

Lines 76-79: Do you have stats or references in South Korea?

Line 88: You also collected samples from Seoul and Busan cities.

Line 94: What kind of methods did you use for the sample size calculation? Is this for the detection of disease? What was the confidence level and the detection level?

Line 98: ‘More samples’ are not clear. Do you have comparison ratios among the seasons?

Line 101: The sentence is not clear. Jang et al. (2020) collected samples from Seoraksan National Park, not the whole area of your study.

Line 111: How did you check the size? Did you take pictures with a ruler on the sampling date?

Line 115: The study area from Lim et al. (2015) was the Odaesan National Park, not the whole area of your study.

Table 1: Seul city: it should be Seoul city; Gyong-buk-Bonghwa: This should be Bonghwa, Gyeongsangbuk-do

Line 125: The year should be 2015-2017.

Line 131: Hanm: this should be Hanam.

Line 132: Gyong-buk- Bonghwa: this should be Bonghwa, Gyeongsangbuk-do.

Line 201: FigS2 does not have A-B

Line 203: S2A-B Fig: this is FigS3 A-B

Line 240: S3Fig: this is FigS2.

Line 317: This sentence is not clear. Did you build two models for E.coli and STEC? Why did you conduct regression analysis? You did not explain the regression analysis in the Results or Discussion sections.

Line 335: Table 2: what is TSI?

Line 392: Table 3: you didn’t explain what Cov and Ident mean.

Line 394: Gyong-buk- Bonghwa: this should be Bonghwa, Gyeongsangbuk-do.

Line 406-407: You didn’t explain what NJ and ML mean.

Line 417: Fig 3 is about stx1, not stx1 + stx2

Line 472-474: Table 3 talks about BHG, BUS, CCG, HMS, SKG, and TBG.

Line 486-488: Is this about table 1? Did you talk about the statistical test in the Materials & Methods section: What kind of test did you use? There’s no explanation of this test in MM (lines 317-324).

Lines 489-490: This sentence is confusing. You can write that the percentage of positive samples was the highest in spring, followed by summer and fall.

Line 504: This is the S3 Table, not S2 Table. This sentence is confusing. Oct and Jul had the highest percentage and number of stx1 gene, but spring has the highest prevalence of stx1. You have to explain it.

Line 510: Fig 6. It is recommended to delete winter because you didn’t collect during winter. You need the number of samples per bar.

Line 520: Add references.

Line532: It is not clear why you add Ju et al. (2011).

Line 535: It will be better to add that Jay et al. (2007) collected samples from California.

Line 539-544: It is not clear what the authors try to say. Why did you add this sentence? Are you trying to say that deer feces is not important because of the low level of E.coli O157 from deer feces with no cross-contamination from infected livestock?

Line 560: Add references.

Lines 561-563: Why do you jump to GAP? Do Korean GAPs include control of wildlife? Do you want to discuss how Korean GAPs contribute to the improvement of food safety?

Lines 586-594: What is Fig FB? What does the high genetic diversity among Korean Salmonella strains mean?

Lines 604-606: You found STEC and Salmonella from wildlife feces, BUT you didn’t prove that wildlife samples are a source of fecal-borne pathogen dissemination. Is there any chance that the irrigation water contaminated with livestock feces transmits pathogenic bacteria?

Reviewer #2: S2A-B Fig. Please also use the italics style for bacterial designation and genes

line 237: β-lactum typos need to be corrected

272: serovars need to be given with an initial capital letter and not in italics

Please also check the references section for the use of the italics style for bacterial and gene designation

Nice work

Reviewer #3: Park et al. present a very large survey of foodborne bacteria presence in wildlife feces throughout S. Korea. This is a valuable sample set with the potential for impactful conclusions to be made regarding produce safety and sampling. Unfortunately, the rich ecological data collected is not presented in a meaningful way. Rather than simply giving prevalence for different cities, I'd like to see if patterns were detected regarding proximity of the positive specimens to the produce - were they at the edges of the field? Or interspersed with the plantings? Near water sources? Also, what is more problematic are the inferences about diversity based on partial gene Sanger sequencing. This is totally antiquated and not scientifically justified. Phylogeography and source tracking of bacteria should be based on whole genome sequencing. This is widely commercially available now for not much more than the cost of PCR.

Data availability: The indicated NCBI numbers are not showing up when I search for them.

Line 117: Were all samples confirmed with the cameras? If not, what proportion?

Line 93: On what assumptions was the sample size calculation based on?

Table 1: Clarify what numbers are given in the "No. of Salmonella detected sample" column

Line 147: Colony morphology is not a confirmatory test

6. PLOS authors have the option to publish the peer review history of their article (what does this mean?). If published, this will include your full peer review and any attached files.

Reviewer #1: No

Reviewer #2: No

Reviewer #3: No

---

## [Author Response · Author response to Decision Letter 0]

5 Dec 2022

Academic editor comments and responses

Comment 1. Please ensure that your manuscript meets PLOS ONE's style requirements, including those for file naming. The PLOS ONE style templates can be found at 

Response 1: we followed the journal guideline and journal requirements

Comment 2. To comply with PLOS ONE submissions requirements, in your Methods section, please provide additional information regarding the experiments involving animals and ensure you have included details on (1) methods of sacrifice, (2) methods of anesthesia and/or analgesia, and (3) efforts to alleviate suffering.

Response 2: We mention ethical issues in our Methods section, no sacrifice animals or anesthesia and/or analgesia.

Comment 3.

As part of your revision, please complete and submit a copy of the Full ARRIVE 2.0 Guidelines checklist, a document that aims to improve experimental reporting and reproducibility of animal studies for purposes of post-publication data analysis and reproducibility: https://arriveguidelines.org/sites/arrive/files/documents/Author%20Checklist%20-%20Full.pdf Please include your completed checklist as a Supporting Information file. Note that if your paper is accepted for publication, this checklist will be published as part of your article.

Response 3: Supporting Information file are attached

Comment 4.

In your Methods section, please provide additional information regarding the permits you obtained for the work. Please ensure you have included the full name of the authority that approved the field site access and, if no permits were required, a brief statement explaining why.

Response 4: permission already mention in methods section, field site there is no need any permission from field sites due to any damage or situation

Comment 5.

We note that the grant information you provided in the ‘Funding Information’ and ‘Financial Disclosure’ sections do not match. 

Response 5: We make a mistake for mentioning the Funding Information’ and ‘Financial Disclosure’. Now we make correct and online mention funding information.

Comment 6.

Response 6: Acknowledgments 

Comment 7. PLOS requires an ORCID iD for the corresponding author in Editorial Manager on papers submitted after December 6th, 2016.

Response 7 : updated

Comment 8.

8. We note that you have included the phrase “data not shown” in your manuscript. Unfortunately, this does not meet our data sharing requirements. PLOS does not permit references to inaccessible data. We require that authors provide all relevant data within the paper, Supporting Information files, or in an acceptable, public repository. Please add a citation to support this phrase or upload the data that corresponds with these findings to a stable repository (such as Figshare or Dryad) and provide and URLs, DOIs, or accession numbers that may be used to access these data. Or, if the data are not a core part of the research being presented in your study, we ask that you remove the phrase that refers to these data.

Response 8: Updated and deleted the “data not shown”

Comment 9. our ethics statement should only appear in the Methods section of your manuscript. If your ethics statement is written in any section besides the Methods, please move it to the Methods section and delete it from any other section. Please ensure that your ethics statement is included in your manuscript, as the ethics statement entered into the online submission form will not be published alongside your manuscript. 

Response 9. Updated

Comment 10. 10. We note that Figure 1 in your submission contain [map/satellite] images which may be copyrighted. All PLOS content is published under the Creative Commons Attribution License (CC BY 4.0), which means that the manuscript, images, and Supporting Information files will be freely available online, and any third party is permitted to access, download, copy, distribute, and use these materials in any way, even commercially, with proper attribution. For these reasons, we cannot publish previously copyrighted maps or satellite images created using proprietary data, such as Google software (Google Maps, Street View, and Earth). For more information, see our copyright guidelines: http://journals.plos.org/plosone/s/licenses-and-copyright.

Response 10. Updated (Fig 1 legend) and attached the copyright permission doc (“ Copy right License (Fig 1).pdf”. 

Reviewer: 1

The manuscript did not thoroughly explain the association between wildlife, agricultural produce, and food poisoning in South Korea. The authors did not fully present the logic or reference whether the food poisoning in South Korea was caused by the consumption of agricultural produce which might be contaminated by wildlife feces. Statistical analysis was not explained comprehensively. This manuscript did not present in an intelligent fashion: more than ten sentences in the Introduction and Discussion sections require adding or changing references. The naming of the collection points was not consistent and a lot of typos and punctuation errors. It is recommended to use a professional copyediting service.

Thanks for your intelligent comments which are important for upgrading our manuscript. 

We apologize for our careless work; we made some collection point and abbreviation errors. A native English speaker reviewed the manuscript and corrected it (attached is an English correction certificate). Lines 78-91, we added extra information related to animals particularly in South Korea. However, there are few direct evidences in South Korea of wild animal feces contaminating agricultural products (ref, 3, 5, 6) but directly no such evidence from wild animals that can contaminate agricultural produce (ref 10, indirect evidence). Therefore, we conducted a wild animal fecal contamination experiment as the most likely source of cross-contamination.

Comment: 1.

Line 50: References are not sufficient. The authors talked about wild animals but references 1-4 were about chickens, food animals, and humans.

Response 1: 

Thanks for your nice observation and comments. We updated the references as you mentioned the wild animal related references. Directly there are very few evidences in South Korea, related to wild animal fecal contaminated to agricultural produces, thus we conducted such kind of experiment of wild animal fecal contamination as we considered the probable source of cross-contamination.

Comment: 2.

Line 51: Add reference for humans.

Response 2: Updated

Comment: 3.

Lines 51-53: The sentence is confusing. Is fecal-oral transmission the only transmission route? Salmonella can spread from animals to people and from people to people.

Response 3: Updated

Comment: 4.

Lines 57-58: Is this for humans or is this include animals? Bai et al. (2016) reported non-O157 STEC.

Response 4: fixed

Comment: 5.

Lines 59-60: Mogren et al (2018) discussed the hurdle approach to control pathogenic bacterial contamination of leafy green vegetables.

Response 5: deleted the reference

Comment: 6.

Line 62: High prevalence is not clear. It is suggested to elaborate on the prevalence and host ubiquity written by Kim et al. (2020)

Response 6: Updated

Comment: 7.

Line 67: Add areas.

Response 7: Updated

Comment: 8

Lines 76-79: Do you have stats or references in South Korea?

Response 8: Updated

Comments: 9.

Line 88: You also collected samples from Seoul and Busan cities.

Response 9: 

We collected samples from a larger area of the capital city of Seoul's nearby agricultural area, as well as a large city of Busan.

Comments: 10.

Line 94: What kind of methods did you use for the sample size calculation? Is this for the detection of disease? What was the confidence level and the detection level?

Response 10: 

The previous experience and reference papers. 

The considered confidence level was 99%, population size unknown, expected standard deviation 0.5, accepted absolute error d = 0.005, and final sample size was 664. (ref link : Glenn 2002; http://www.winepi.net/uk/index.htm)

Comment: 11.

Line 98: ‘More samples’ are not clear. Do you have comparison ratios among the seasons?

Response: 11

 Updated

Comment: 12.

Line 101: The sentence is not clear. Jang et al. (2020) collected samples from Seoraksan National Park, not the whole area of your study.

Response 12: The representative survey area. (Seoraksan National Park in South Korea)

Comment: 13.

Line 111: How did you check the size? Did you take pictures with a ruler on the sampling date?

Response 13: We provided images of fecal sample from different animals. See the supp. FigS2. (comments 2 of reviewer 3)

Comment: 14.

Line 115: The study area from Lim et al. (2015) was the Odaesan National Park, not the whole area of your study.

Response 14: The representative survey area.

Comment: 15.

Table 1: Seul city: it should be Seoul city; Gyong-buk-Bonghwa: This should be Bonghwa, Gyeongsangbuk-do

Response 15: Updated

Comment: 16

Line 125: The year should be 2015-2017.

Response 16: Updated

Comments: 17.

Line 131: Hanm: this should be Hanam.

Response 17: Updated

Comment: 18.

Line 132: Gyong-buk- Bonghwa: this should be Bonghwa, Gyeongsangbuk-do.

Response 18: Updated

Comment: 19.

Line 201: FigS2 does not have A-B

Response 19: Updated

Comment: 20.

Line 203: S2A-B Fig: this is FigS3 A-B

Response 20: Updated

Comment: 21.

Line 240: S3Fig: this is FigS2.

Response 21: Updated

Comments: 22.

Line 317: This sentence is not clear. Did you build two models for E.coli and STEC? Why did you conduct regression analysis? You did not explain the regression analysis in the Results or Discussion sections.

Response 22: Updated (regression analysis was conducted due to relation with seasons and host (wild animal) prevalence of tested pathogens outcome. We built models each of the pathogen and validated model and the odds ratio (OD value) and 95% confidence interval (CI) were derived.

Comment: 23.

Line 335: Table 2: what is TSI?

Response 23: Updated

Comment: 24.

Line 392: Table 3: you didn’t explain what Cov and Ident mean.

Response 24: Updated

Comment: 25

Line 394: Gyong-buk- Bonghwa: this should be Bonghwa, Gyeongsangbuk-do.

Response 25: Updated

Comment: 26

Line 406-407: You didn’t explain what NJ and ML mean.

Response 26: Updated

Comment: 27

Line 417: Fig 3 is about stx1, not stx1 + stx2

Response 27: Updated

Comment: 28

Line 472-474: Table 3 talks about BHG, BUS, CCG, HMS, SKG, and TBG.

Response 28: Updated

Comment: 29

Line 486-488: Is this about table 1? Did you talk about the statistical test in the Materials & Methods section: What kind of test did you use? There’s no explanation of this test in MM (lines 317-324).

Response 29: We mentioned statistical test in the Materials & Methods section and result section. 

Comments: 30

Lines 489-490: This sentence is confusing. You can write that the percentage of positive samples was the highest in spring, followed by summer and fall.

Response 30: Updated as your recommendation

Comment: 31

Line 504: This is the S3 Table, not S2 Table. This sentence is confusing. Oct and Jul had the highest percentage and number of stx1 gene, but spring has the highest prevalence of stx1. You have to explain it.

Response 31: Sorry for careless error. Updated

Comment: 32

Line 510: Fig 6. It is recommended to delete winter because you didn’t collect during winter. You need the number of samples per bar.

Comment: 32

Response: deleted the winter season from Fig 6

Comment: 33

Line 520: Add references.

Response 33: Updated

Comment: 34

Line532: It is not clear why you add Ju et al. (2011).

Response 34: Updated

Comment: 35

Line 535: It will be better to add that Jay et al. (2007) collected samples from California.

Response 35: Added California region as your recommendations

Comment: 36

Line 539-544: It is not clear what the authors try to say. Why did you add this sentence? Are you trying to say that deer feces is not important because of the low level of E. coli O157 from deer feces with no cross-contamination from infected livestock?

Response 36: Updated

Comments: 37

Line 560: Add references.

Response 37: Updated

Comments: 38

Lines 561-563: Why do you jump to GAP? Do Korean GAPs include control of wildlife? Do you want to discuss how Korean GAPs contribute to the improvement of food safety?

Response38: Updated (Line No. (page 28): 619-627)

Comment: 39

Lines 586-594: What is Fig FB? What does the high genetic diversity among Korean Salmonella strains mean?

Response: Updated

Comment:40

Lines 604-606: You found STEC and Salmonella from wildlife feces, BUT you didn’t prove that wildlife samples are a source of fecal-borne pathogen dissemination. Is there any chance that the irrigation water contaminated with livestock feces transmits pathogenic bacteria?

Response 40

Thanks for your observation and thoughtful comment. We agree with you. Wildlife feces may be a source of fecal-borne pathogen dissemination in various areas of South Korea.

Reviewer 2: 

Comment: 1.

#2: S2A-B Fig. Please also use the italics style for bacterial designation and genes

Response: 1. Updated

Comment: 2.

line 237: β-lactum typos need to be corrected

Response: 2. Updated

Comment: 3.

272: serovars need to be given with an initial capital letter and not in italics

Response 3: Updated

Comment:4.

Please also check the references section for the use of the italics style for bacterial and gene designation

Response 4: We carefully checked and updated whole manuscript.

Reviewer 3: comments and responses 

Reviewer #3: Park et al. present a very large survey of foodborne bacteria presence in wildlife feces throughout S. Korea. This is a valuable sample set with the potential for impactful conclusions to be made regarding produce safety and sampling. Unfortunately, the rich ecological data collected is not presented in a meaningful way. Rather than simply giving prevalence for different cities, I'd like to see if patterns were detected regarding proximity of the positive specimens to the produce - were they at the edges of the field? Or interspersed with the plantings? Near water sources? Also, what is more problematic are the inferences about diversity based on partial gene Sanger sequencing. This is totally antiquated and not scientifically justified. Phylogeography and source tracking of bacteria should be based on whole genome sequencing. This is widely commercially available now for not much more than the cost of PCR.

Response: 

Thanks for your positive comments. We agree with your comments rich ecological data collected is not presented in a meaningful way. We collected feces sample nearby agricultural areas, closely water stream, but exactly or systematically we could not point out the prevalence pattern based on the exact ecological consequence. 

In general, if a fecal sample pile is found, we look for feces approximately few hundred meters away from the fecal pile even though animals tend to roam in the nearby region. A single pile of all fecal pellets in one location was counted as one sample. The geographic points of each fecal scat were recorded using GPS co-ordination. We collected fecal samples nearby valleys, streams, farmlands, and reservoirs adjacent to the target agricultural area). Currently, we cannot relate the positions of positive specimens to habitat types, thus, just we can provide the fig-survey areas and collection of samples- and sample collecting place belong to forest and watery areas near agricultural area including agricultural land in Materials and Method. For your kind consideration we provided/ added two figures S1 and S3.

Comment: 1

Data availability: The indicated NCBI numbers are not showing up when I search for them.

Response: 1

The accession numbers have already been submitted, but the accession numbers can be followed to NCBI after publication/provided date with authors (restricted by the author).

Comment: 2

Line 117: Were all samples confirmed with the cameras? If not, what proportion?

Response: 2 

We only used the trail camera to survey the animal movement and not to confirm the samples. During the collection period, the samples were examined for morphology, color, shape, and size using a scale (attached the feces color, shape fig S2 in the material and methods).

Comment: 3

On what assumptions was the sample size calculation based on?

Response: 3

Thanks for your comment. Sample size calculations-based o the previous experience, published literature. We follow the following link: http://www.winepi.net/uk/sample/indice.htm

the considered confidence level was 99%, population size unknown, expected standard deviation 0.5, accepted absolute error d = 0.005, and final sample size was 664. We then aimed to collect approximately 700 samples within three seasons: spring (March–May), summer (June–August), and autumn (September–November); the following website calculations were helped http://www.winepi.net/uk/sample/indice.htm

Comment: 4

Table 1: Clarify what numbers are given in the "No. of Salmonella detected sample" column

Response: Updated 

Line 147: Colony morphology is not a confirmatory test

Response: Updated

---

## [Decision Letter · Decision Letter 1]

13 Jan 2023

Prevalence of Salmonella spp. and Escherichia coli in the feces of free-roaming wildlife throughout South Korea

PONE-D-22-23426R1

Dear Dr. Park,

We’re pleased to inform you that your manuscript has been judged scientifically suitable for publication and will be formally accepted for publication once it meets all outstanding technical requirements.

Kind regards,

Csaba Varga, DVM MSc PhD

Academic Editor

PLOS ONE

Additional Editor Comments (optional):

Thank you for addressing all of the comments!

I have carefully reviewed the manuscript and the authors addressed all of the peer-reviewers' suggestions. Reviewer 3 rejected the manuscript, however the  whole-genome sequencing analysis that she suggested should be a future research direction and not a requirement for publication of this current study.

The article as currently stands qualifies for publication based on PLOS ONE publication criteria.

Reviewers' comments:

Reviewer's Responses to Questions

**Comments to the Author**

1. If the authors have adequately addressed your comments raised in a previous round of review and you feel that this manuscript is now acceptable for publication, you may indicate that here to bypass the “Comments to the Author” section, enter your conflict of interest statement in the “Confidential to Editor” section, and submit your "Accept" recommendation.

Reviewer #2: All comments have been addressed

Reviewer #3: (No Response)

2. Is the manuscript technically sound, and do the data support the conclusions?

Reviewer #2: Yes

Reviewer #3: No

3. Has the statistical analysis been performed appropriately and rigorously? 

Reviewer #2: N/A

Reviewer #3: Yes

4. Have the authors made all data underlying the findings in their manuscript fully available?

Reviewer #2: Yes

Reviewer #3: No

5. Is the manuscript presented in an intelligible fashion and written in standard English?

Reviewer #2: Yes

Reviewer #3: Yes

6. Review Comments to the Author

Reviewer #2: The revision have been carefully addressed. There are no further comments of this reviewer to improve the quality of the manuscript.

Reviewer #3: I am not satisfied with the revisions, in particular for my comment on use of partial gene sequence for Salmonella typing, which was essentially ignored. The paper is still quite disorganized.

7. PLOS authors have the option to publish the peer review history of their article (what does this mean?). If published, this will include your full peer review and any attached files.

Reviewer #2: No

Reviewer #3: No

---

## [Editor Report · Acceptance letter]

3 Apr 2023

PONE-D-22-23426R1 

Prevalence of *Salmonella* spp. and *Escherichia coli* in the feces of free-roaming wildlife throughout South Korea 

Dear Dr. Chul:

I'm pleased to inform you that your manuscript has been deemed suitable for publication in PLOS ONE. Congratulations! Your manuscript is now with our production department. 

Kind regards, 

on behalf of

Dr. Csaba Varga 

Academic Editor

PLOS ONE